# Decoding arm speed during reaching

Yoh Inoue[1], Hongwei Mao[2,3], Steven B. Suway[3,4], Josue Orellana[3,5] & Andrew B. Schwartz[2,3,6]

Neural prostheses decode intention from cortical activity to restore upper extremity movement. Typical decoding algorithms extract velocity—a vector quantity with direction and magnitude (speed) —from neuronal firing rates. Standard decoding algorithms accurately recover arm direction, but the extraction of speed has proven more difficult. We show that this difficulty is due to the way speed is encoded by individual neurons and demonstrate how standard encoding-decoding procedures produce characteristic errors. These problems are addressed using alternative brain–computer interface (BCI) algorithms that accommodate nonlinear encoding of speed and direction. Our BCI approach leads to skillful control of both direction and speed as demonstrated by stereotypic bell-shaped speed profiles, straight trajectories, and steady cursor positions before and after the movement.

[1] Department of Neurosurgery, Osaka University Graduate School of Medicine, Osaka 565-0871, Japan. [2] Systems Neuroscience Center, University of Pittsburgh, Pittsburgh, PA 15213, USA. [3] Center for the Neural Basis of Cognition, Carnegie Mellon University and University of Pittsburgh, Pittsburgh, PA 15213, USA. [4] Center for Neuroscience, University of Pittsburgh, Pittsburgh, PA 15260, USA. [5] Machine Learning Department, Carnegie Mellon University, Pittsburgh, PA 15213, USA. [6] Department of Neurobiology, University of Pittsburgh, Pittsburgh, PA 15260, USA. Correspondence and requests for materials should be addressed to A.B.S. (email: abs21@pitt.edu)

The firing rates of individual motor cortical neurons encode the direction[1,2] and speed[3,4] of arm movement. This encoded information is the basis for neural prostheses used by paralyzed subjects to regain lost arm and hand movement[5–9]. Although the demonstrated performance in displacing the arm and shaping the hand while using these prosthetic devices approximates that of normal subjects[10], problems remain at the end of a reach as a target is acquired[11–13]. Decoders used for intracortical neural prosthetic arm movement estimate subjects' intended velocity vectors. The directions of these vectors are quite accurate, but their magnitudes (speed) are often problematic[14]. Neuronal encoding of kinematic parameters can be modeled using tuning functions, and one common tuning equation[3] (Eq. 5 below) describes the effect of speed on firing rate in two ways: (1) as a gain on the cosine tuning function for direction, and (2) as a speed-dependent offset. The gain effect is largest when moving along the neuron's preferred direction (direction for which the neuron fires maximally). Typical decoders work by inverting encoding models fit to the recorded firing rates. However, this speed-direction model, with speed having an effect on both gain and offset, cannot be inverted directly. Instead, encoding equations with only an operational gain term are typically used for velocity decoding. Interestingly, since the classic population vector is constructed with empirical firing rates, this speed-direction interaction is captured as a change in the length of the resultant vector, even though the "direction-only" encoding model used in this decoder has no explicit terms for speed. As a result, the length of the population vector reflects movement speed[1]. However, because the population vector algorithm and other encoder-based decoders fail to account for all the speed-related effects on firing rates, the extracted movement signal may have deviations from the intended movement. In particular, the additive offset term in the encoding model can have a specific effect on decoding performance. This problem could be addressed by decoding algorithms that reduce the effect of variance due to signal residuals or that are capable of handling nonlinear interactions, such as those due to the offset term. In this study, we examine models that describe velocity encoding in individual neurons and how the gain and additive speed factors affect velocity extraction by population decoders. Activity is simulated using different forms of these models and this analysis shows that the speed-offset factor influences subsequent decoding. The same effect is observed when activity from empirically recorded units is decoded. We present alternative formulations that counteract this problem, and demonstrate their effectiveness during normal reaching and brain-controlled virtual reaching.

## Results

**Behavior.** Two monkeys performed center-out movements to 16 targets. Figure 1a shows the average hand trajectories to each target for Monkeys N and C. Figure 1b shows the average speed profile across all targets. The mean (±SD) peak speed for Monkey N was $35.3 \pm 4.93$ cm s$^{-1}$, and $47.6 \pm 5.51$ cm s$^{-1}$ for Monkey C. Both monkeys made straight, accurate reaches with bell-shaped speed profiles.

**Encoders.** The original directional-tuning model for an individual unit proposed by Georgopoulos and colleagues[2] was expressed in the following equation for 2-dimensional movement.

$$y = b_0 + b_x d_x + b_y d_y + \varepsilon \qquad (1)$$

where $y$ is the mean firing rate of the neuron (during the movement), $d_x$ and $d_y$ specify the direction of a target from the initial position of the hand at the beginning of the reach, $b_x$, $b_y$, $b_0$ are regression coefficients, and $\varepsilon$ is the noise (or error)

representing the deviation from the model. This encoding equation was modified[15,16] to incorporate time:

$$y(t - \tau) = b_0 + b_x d_x(t) + b_y d_y(t) + \varepsilon(t) \qquad (2)$$

where $y(t)$ is the firing rate of the neuron at time $t$, and $\tau$ is the time lag between cortical activity and hand direction. We refer to Eq. (2) as the "direction-only model." This formulation can be considered as the inner product between two vectors, $\mathbf{b} = [b_x, b_y]$ lying in the preferred direction (PD), $atan(b_y/b_x)$, and $\mathbf{d}(t) = [d_x(t), d_y(t)]$, a unit vector in the instantaneous direction of movement. Equation (2) can be rewritten to emphasize its directional nature:

$$y(t - \tau) = b_0 + m\cos(\theta(t) - \theta_{PD}) + \varepsilon(t) \qquad (3)$$

where $m = \sqrt{b_x^2 + b_y^2}$ is the modulation depth, $\theta(t)$ is the movement direction at time $t$, and the unit's preferred direction is $\theta_{PD}$.

It was later found[3] that movement speed acts as a gain factor in the encoding of firing rate and that, additionally, speed has an offset effect on the overall tuning function. These two effects are captured in the following equation:

$$y(t - \tau) = b_0 + b_x v_x(t) + b_y v_y(t) + b_s |\mathbf{v}(t)| + \varepsilon(t) \qquad (4)$$

which we refer to as the "offset model." Here, $\mathbf{v}(t)$ is a velocity vector with magnitude $|\mathbf{v}(t)|$, equal to the speed in the instantaneous direction of movement. An equivalent re-parameterization of Eq. (4) is:

$$y(t - \tau) = b_0 + m|\mathbf{v}(t)|\cos(\theta(t) - \theta_{PD}) + b_s |\mathbf{v}(t)| + \varepsilon(t) \qquad (5)$$

A version of this model excludes the offset term[17]:
$y(t - \tau) = b_0 + b_x v_x(t) + b_y v_y(t) + e(t)$ and we refer to this as the "gain-only" model which can be stated equivalently as:

$$y(t - \tau) = b_0 + m|\mathbf{v}(t)|\cos(\theta(t) - \theta_{PD}) + \varepsilon(t) \qquad (6)$$

Note that in these models, the coefficients $b_x$ and $b_y$, and when present, $b_s$, incorporate the combined effects of speed and direction on firing rate. In the subsequent analyses, we show how inclusion of the speed offset term, $b_s$, changes the nature of the encoding model. To evaluate the effect of the offset coefficient relative to the modulation depth, we used the following *offset ratio*:

$$H = \frac{b_s}{m + |b_s|} \qquad (7)$$

**Data.** We first applied the offset encoding model (Eq. 4) to units recorded from the primary motor cortex of monkeys during center-out arm reaching movements. The optimal time lag ($\tau$) between kinematics and a unit's firing (both were trial-averaged) was found based on the coefficient of determination ($R^2$) of models fit at different lags, ranging from $-120$ ms to 270 ms at 30 ms steps. The median lag for the highest R$^2$ values were 90 ms (Monkey N) and 60 ms (Monkey C) (Fig. 2a). The offset model explained much of the variance in our data (Fig. 2b) with median R$^2$s of 0.59 and 0.62, for Monkeys N and C, respectively. To evaluate the importance of the speed-offset term relative to directional modulation for each unit's tuning function, we computed the offset ratio (Eq. 7). A small offset ratio means that a unit's tuning modulation is dominated by direction, whereas a large ratio indicates the effect of speed-offset term is greater than that of direction. The speed offset, $b_s$, was substantial (Fig. 2c) in

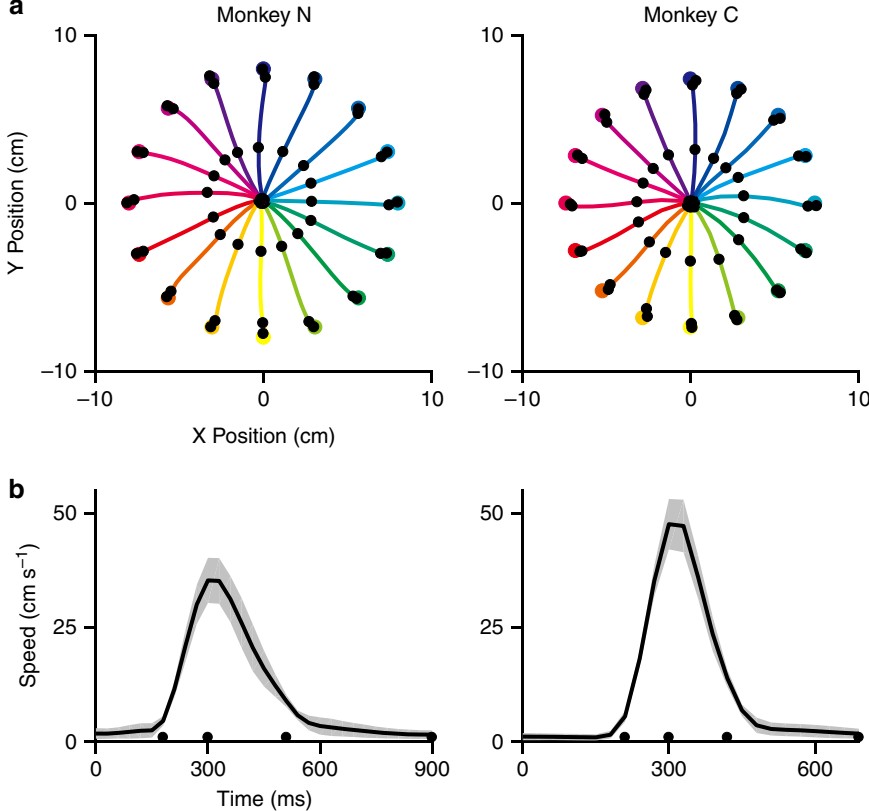

**Fig. 1** Kinematic profiles of Monkeys N and C in the center-out reaching task. The four black dots indicate: movement onset, peak velocity, movement offset, and hold off (this marking applies to all figures). **a** Average hand trajectories, each colored trace represents a target direction (16 total). This colormap also applies to the subsequent figures. **b** Speed profiles pooled across all target directions with the average shown in black bold and standard deviation as the gray shaded region ($n = 69 \times 16$ trials for Monkey N and $47 \times 16$ trials for Monkey C)

the collected data, as the median of the offset ratios calculated at the optimal time lags were 0.57 (Monkey N) and 0.41 (Monkey C). This shows that the speed-offset term is an important factor in describing the firing rate modulation of these units during the center-out task.

To further illustrate the effect of the speed-offset term, we chose three units with different offset ratios. Firing rates from a unit with a small offset ratio (−0.082) are illustrated in Fig. 3a. The traces in this figure show a clear directional effect with a balance of positive and negative modulation, corresponding to movements in the preferred and anti-preferred directions. In contrast, an example unit with a large offset ratio (.868) has firing-rate profiles that are only positive-going and are almost the same for each direction of movement (Fig. 3b). An intermediate ratio (close to 0.5) is indicative of a more equitable effect of speed and direction on firing rate; one such example is the unit (offset ratio = 0.370) shown in Fig. 3c. Here, there is a direction-related modulation of firing rates for the targets within 90 degrees of the preferred direction. However, the profiles in the anti-preferred directions are small and overlap. The distribution of offset ratios in Fig. 2c suggests that most units fall in this intermediate range.

**Simulation**. We used simulations (see Methods) to explore situations in which there was a mismatch between the model used to generate firing rates and the observation model used to describe the way those rates were encoded. Direction-specific temporal patterns of firing rate were investigated using the direction-only encoding model (Eq. 3) when the firing rates were generated with either the gain-only model (Eq. 6) or the offset model (Eq. 5). The left column in Fig. 4 shows simulated firing

rates generated using the gain-only model and velocities from Monkey N's movements. Poisson noise with a rate parameter given by Eq. (8) was included in the simulation. Figure 4a-left shows simulated firing rates (PD = 90°) modulated symmetrically as a function of time about the baseline of 30 Hz. Subsequent regression (with data from the entire course of the trial) using the direction-only encoding model on this simulated neuron yielded:

$$\hat{y} = 30.0 + 4.77 \times \cos(\theta - 90.2°),$$

accurately reflecting the generative process.

The normalized version of firing rate (Eq. 10) is shown in Fig. 4b-left and is symmetrical about zero. Unit-specific contributions to a decoder such as the Population Vector Algorithm (PVA, see Methods) can be represented as a vector oriented along the axis of the unit's preferred direction with a length proportional to its normalized firing rate. The time series of these contributions for a single movement in the preferred direction is shown in Fig. 4c-left. The profile, traced out by the vector tips, matches the speed profile. Note that when movement is in a direction 90 degrees from the preferred direction, the profile is flat and there is no contribution to the population vector (Fig. 4d-left). For movement in the anti-preferred direction (Fig. 4e-left), the firing rates are below the baseline offset and negative. Since negative vectors point in the opposite direction, these vectors also reflect accurate speed coding.

As a comparison to the gain-only model, the column on the right side of Fig. 4 shows a simulation generated with the offset model using velocity taken from Monkey N with Poisson noise and a rate parameter given by Eq. (9). This equation specifies a

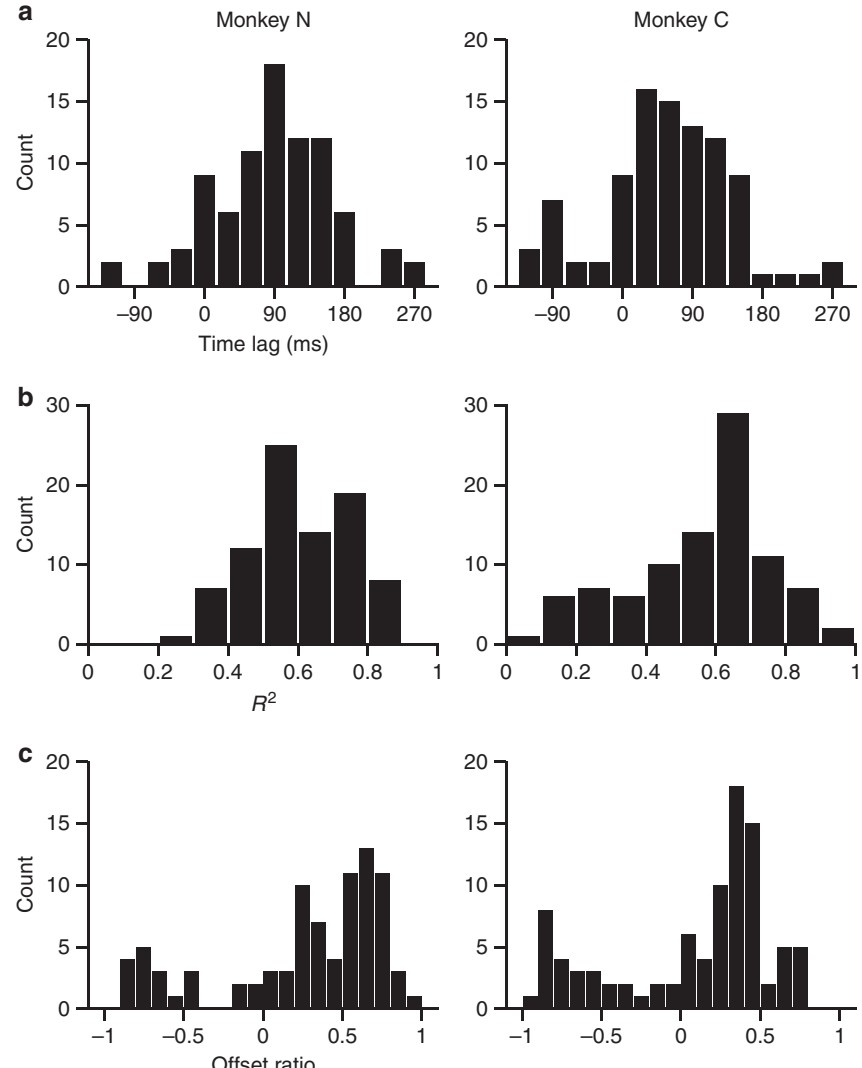

**Fig. 2** Results of regression analysis for Monkey N (left column, 86 units) and Monkey C (right column, 93 units). For each monkey, the histograms in this figure summarize the results from all recorded units. **a** Histogram of the optimal time lags at which $R^2$ value is the highest among the sliding time window: −120 to 270 ms. **b** Histogram of maximum $R^2$ values. **c** Histogram of offset ratios calculated at the optimal time lags

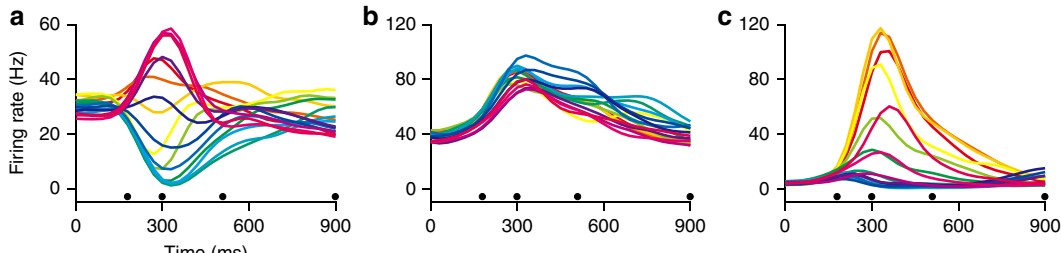

**Fig. 3** Average firing rates of three representative units. Each trace corresponds to a target direction (same colormap as Fig. 1). **a** An example unit (Monkey N, 89-1) with a relatively small offset ratio (−0.082). **b** A unit (Monkey N, 32-2) with a large offset ratio of 0.868. **c** A unit (Monkey N, 55-1) with an intermediate offset ratio of 0.370

value of 0.25 Hz per m s$^{-1}$ for both the offset term and the modulation depth term, yielding an offset ratio of 0.5. Figure 4a-right shows that simulated firing rates with PD = 90°. Here too, the firing rates are modulated as a function of time, but the profiles to different targets have amplitudes that are asymmetric

about the baseline of 30 Hz. If the direction-only model is used to fit this simulated neuron, the resulting equation is:

$$\hat{y} = 32.5 + 2.41 \times \cos(\theta - 89.9°)$$

Notice that the estimated offset term, $b_0$, is now 32.5 instead of 30 impulses per second. The misestimated offset is evident as

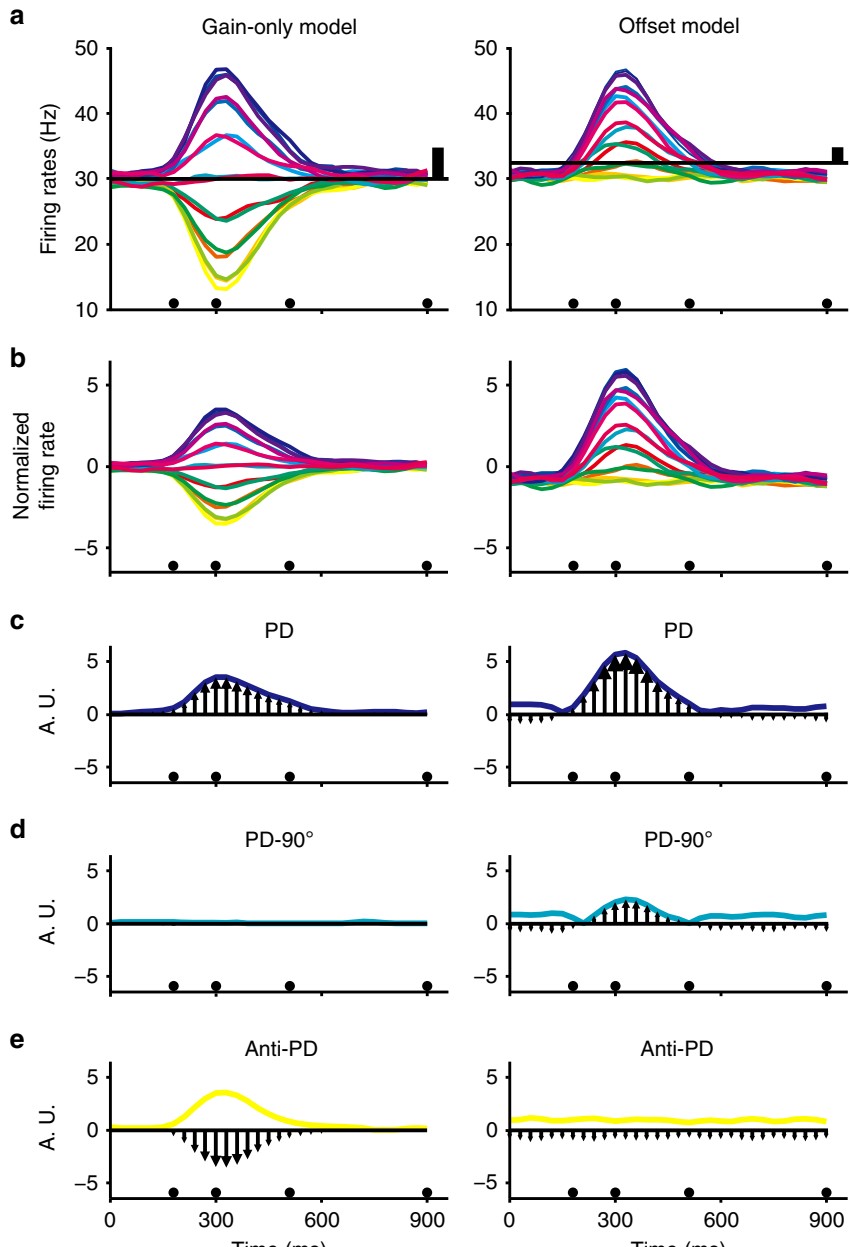

**Fig. 4** The use of a conventional direction-only encoding model when firing rates were generated with either the gain-only (left column) or offset model (right column). **a** Simulated firing rates for one neuron and 16 target directions (same color map as in Fig. 1). The horizontal black lines represent estimated baseline, vertical black lines represent estimated modulation depth. **b** Firing rates normalized by estimated baseline and modulation depth using Eq. (10). **c** Vector magnitudes pointing to the preferred direction $\theta_{PD}$ (arrows), the colored traces represent the absolute magnitudes. **d** Same as **c** for the orthogonal direction to $\theta_{PD}$, ($\theta_{PD}$ - 90°). **e** Same as **c** in the anti-preferred direction ($\theta_{PD}$-180°). A.U.: arbitrary unit

baseline firing rates, before and after movement, that are now below the estimated baseline. Since direction-only models are often assumed when constructing decoders, the presence of an offset effect in actual data would be expected to produce similar errors. Because of this baseline error, the normalized firing rates to all 16 targets started and ended with negative values after normalization (Fig. 4b-right).

For a single movement in the preferred direction (Fig. 4c-right), the time series of contributions before and after the movement are also negative, pointing in the anti-preferred direction. Instead of a zero contribution before and after the movement (e.g., the gain-only model on the left), these vectors tend to add to the magnitude

of the decoded output. When movements are made at a 90-degree angle from the preferred direction (Fig. 4d-right), instead of making a near-zero rate as expected for the direction-only model, units following the offset model contribute to the decoded speed throughout the movement. The deleterious effect of offset encoding is further illustrated for movement in the anti-preferred direction. The profile is now flat and does not reflect the speed profile (Fig. 4e-right).

As a consequence of using a direction-only encoding model on data that follow the offset-tuning equation, the constant, $b_0$, will be misestimated in the regression. When this constant is then used in Eq. (10) to normalize firing rates, the decoder will

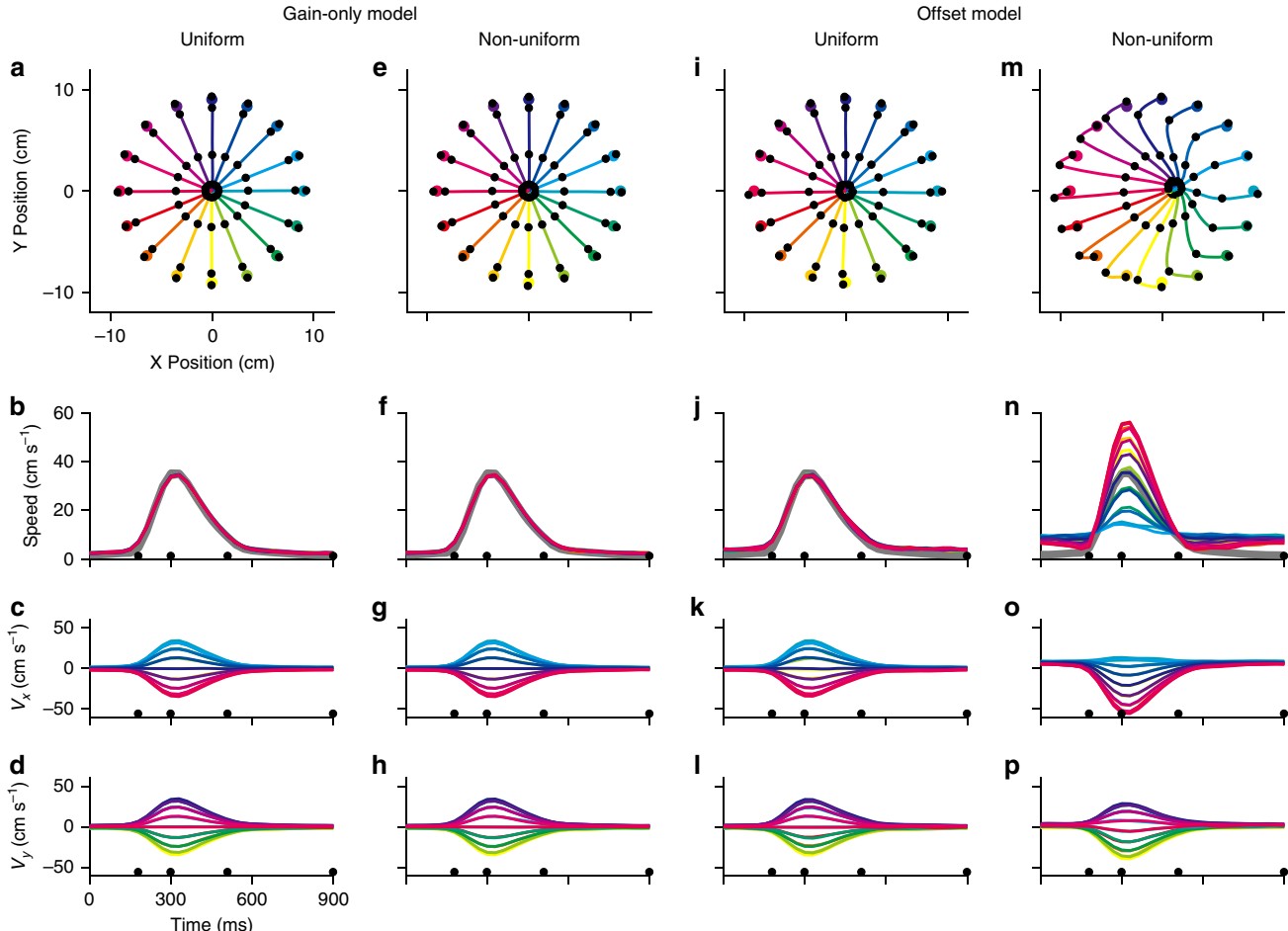

**Fig. 5** Population decoding. Population decoding (using minimal OLE) in four simulated conditions: uniform (**a–d**) and non-uniform (**e–h**) distribution of preferred directions using the gain-only model, and uniform (**i–l**) and non-uniform (**m–p**) distribution of preferred directions with the offset model. **a**, **e**, **i**, **m** reconstructed trajectories for the 16 targets. **b**, **f**, **j**, **n** reconstructed speed profiles. The thick gray line represents the original speed profile (which also applies to subsequent figures). The decoded velocity components, $V_x$ and $V_y$, are shown respectively in **c**, **g**, **k**, **o**; **d**, **h**, **l**, **p**

incorrectly have a non-zero speed representation before and after the movement. This is commonly observed as "drift" or decoded movement occurring when there should not be any. One way to rectify this problem in BCI applications would be to use a spontaneous firing rate, (e.g., from a pre-movement hold period) as an empirically-derived value of $b_0$. While this may work for controlled paradigms such as the center-out task, in practical use with spontaneous and novel movement, this may not be possible. In addition to this problem, non-zero contributions for movements in the directions orthogonal to the preferred direction, as well as the absence of contributions in the anti-preferred direction, are likely (depending on the sample of preferred directions) to distort the resultant decoded direction.

**Decoders**. We can show how standard decoders perform by simulating firing rates using the gain-only and offset speed encoding models. These firing rates were applied to several population decoders (see Methods): the standard, 'minimal' Optimal Linear Estimator (OLE, Eq. 12, 13)[18], Direct Regression (Eq. 15, 16), and an artificial neural network (ANN, Eq. 17, 18). The results show how the standard decoding may be affected by failing to account for firing rates that follow the offset model and how alternative decoders may compensate for this problem. These decoders were then used to reconstruct center-out arm movements from recorded neural activities. Decoder performance

was further tested during online closed-loop BCI control of virtual reaching movements.

**Decoding with simulated neural units**. In order to explore the conditions likely to cause errors in the decoding of velocity, we simulated center-out firing rates for 36 units in four conditions (Fig. 5): (1) a uniform distribution of preferred directions with the gain-only model; (2) a non-uniform distribution (more units with preferred directions in the range of 90 to 270 degrees, see Methods) with the gain-only model; (3) a uniform distribution of preferred directions using the offset model; and, (4) a non-uniform distribution of offset-model firing rates.

Movement trajectories were decoded using population vectors derived from the minimal OLE decoder (which operated on firing rates using the direction-only encoding model). The first three conditions (uniform and non-uniform preferred direction distributions generated with the gain-only model, and uniform preferred direction distribution under the offset model, Fig. 5a–l) matched the simulated movements well. The decoded trajectories were straight and accurate, the speed profiles matched the simulated speeds consistently to each target, and the component profiles of the contributory vectors accurately reflected the X and Y components of the simulated velocity vectors. However, in the last condition (non-uniform preferred direction distribution under the offset model, Fig. 5m–p), the OLE decoding was

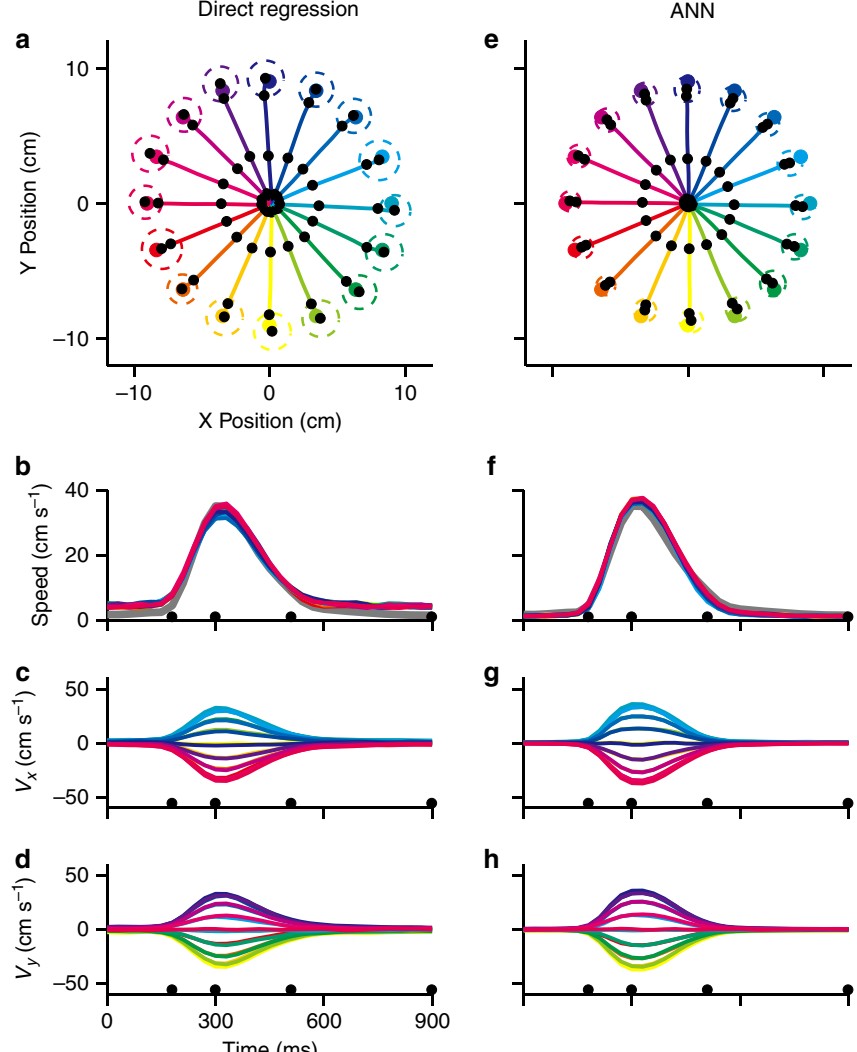

**Fig. 6** Population decoding using data simulated with non-uniform distribution of preferred directions and the offset model. **a**–**d** Direct Regression decoder. **e**–**h** ANN decoder. **a**, **e** reconstructed trajectories for the 16 targets. The radii of the dashed circles represent the averaged distance between end-point positions and their mean. **b**, **f** reconstructed speed profiles. The thick gray line represents the original speed profile. The decoded velocity components, $V_x$ and $V_y$, are shown respectively in **c**, **g**; **d**, **h**

inaccurate. The extracted trajectories overshot the left targets and the trajectories were not straight. Speed profiles in this condition had direction-dependent amplitudes (Fig. 5n). The $X$ components of the predicted velocity (Fig. 5o) were under-modulated for the right-side targets (+90 to −90 degrees) and over-modulated for the left targets (90 to 270 degrees), while the profiles of the $Y$ components (Fig. 5p) were less distorted. The speed-offset effect is evident as directionally-biased distortion when the preferred direction sample is non-uniform.

The standard decoding procedure of inverting a population of encoding models (as used by PVA and OLE) can be bypassed altogether by finding the weights of each unit's contribution to the population that yield the best match to the actual velocity. Results from a linear form of this decoder, Direct Regression, on the simulated firing rates (non-uniform distribution of preferred directions and the offset model, as used in Fig. 5m–p) are shown in Fig. 6a–d. The decoded trajectories were straight and accurate and the speed profiles were bell-shaped and consistent across directions.

An alternative decoder, the ANN, like the Direct Regression method, finds an optimized way to combine each neuron's contribution. The ANN also has the advantage of converging to a

solution when the mapping from input (firing rates) to output (velocities) is nonlinear. Trajectories resulting from this decoder were also straight and accurate with bell-shaped velocity profiles (Fig. 6e–h). Because we included Poisson noise when constructing the simulated firing rates, there was variation in the decoded trajectories. To quantify this variation, we calculated the distance between end-point position of each trajectory and the trial-averaged end position for each target. The median of this distance across all trials ($n = 16 \times 50 \times 10$) was 1.21 cm and 0.80 cm for the Direct Regression and ANN, respectively. The trajectory variability of the ANN was smaller (Mann–Whitney $U$-test, $p < 0.001$, Bonferroni).

These same results can be demonstrated analytically by constructing population vectors with a description of the resulting directional accuracy and speed offset. The different generative models (gain-only, offset) were again used to simulate firing rates and these were then regressed with the standard equation (Eq. 1) to find $\hat{b}_x, \hat{b}_y, \hat{b}_0$ for each simulated unit. The overall equation for the $X$-component of the population vector when using the offset model can be shown to be (Supplementary Methods):

$$\hat{v}_x(t) = \hat{b}_{0P} + \left|\hat{\mathbf{b}}_{P_x}\right| \cdot |\mathbf{v}(t)| \cdot \cos\left(\theta(t) - \hat{\theta}\right) + \hat{b}_{sP}|\mathbf{v}(t)| \ \text{where:}$$

$\left|\hat{\mathbf{b}}_{P_x}\right|$ is the coefficient of directional tuning on the population

$|\mathbf{v}(t)|$ is the speed

$\theta$ is the movement direction (specified)

$\hat{\theta}$ is the predicted angle of the $X$ axis (e.g., 0°)

$\hat{b}_{sP}$ is the effect of the speed offset on the population

$\hat{b}_{0P}$ is the constant offset for the population

The results of the simulation-based decoders for $X$ and $Y$ are shown in Supplementary Table 1. Using the minimal OLE on data generated with an offset model and a non-uniform distribution of preferred directions, resulted in a speed offset term $\left(\hat{b}_{sP}\right)$ that was four times larger for the $X$ component than for $Y$. The baseline offset, $\hat{b}_{0P}$, was large and biased in the $X$ dimension. This caused a skewing in the decoded trajectory components as evidenced in the result shown in Fig. 5o, p. In contrast, the speed offset coefficient for the Direct Regression decoder was very small, as was that for the baseline offset as visualized by the symmetric profiles in Fig. 6c, d. The Direct Regression decoder can compensate for the same non-uniform distribution of data generated from the offset model. The ANN decoder must be similarly effective (Fig. 6e–h) although analytical results are not available due to the nonlinear operations it employs.

**Performance with arm movement data**. We tested the minimal OLE, Direct Regression, and ANN decoders with arm movement data. For the OLE decoder, tuning parameters were found using the direction-only regression model (Eq. 2; without trial averaging). Only units with an $R^2 > 0.03$ (33 units for Monkey N and 19 units for Monkey C) were used with all decoders to predict movement trajectories. The decoding was carried out separately for each monkey (Fig. 7). Trajectories decoded with OLE were skewed and the speed profiles had variable amplitudes for movements to different targets (Fig. 7c, d for Monkey N, Fig. 7k, l for Monkey C). The Direct Regression yielded modest improvements in accuracy (Fig. 7e, f, m, n). The ANN produced accurate representations of the trajectories (correlation coefficient $r = 0.91$ for $V_x$, 0.89 for $V_y$ − Monkey N; $r = 0.84$ for $V_x$, 0.86 for $V_y$ − Monkey C) for both monkeys (Fig. 7g, o). Speed profiles were consistent between targets and matched the arm speed closely (Fig. 7h, p). The reconstructed trajectories and speed profiles were represented more precisely for both monkeys. Vector field correlations[19] between ANN estimates and real arm movement velocities for both monkeys were significantly higher than those calculated using any other decoder in each target direction ($p < 0.001$, $t$-test, Bonferroni) (Fig. 7b, j).

**Performance with BCI virtual arm movement**. The performance of the OLE, Direct Regression and ANN decoders for closed-loop movement control was further tested with Monkey N using a brain–computer interface (BCI). The variance-only OLE (see Methods) was used instead of the minimal OLE to address neuron-specific variance related to speed. OLE trajectories were slightly skewed, but straighter than those decoded from the offline data (Fig. 7c). In addition to the difference between the "minimal" and "variance-only" versions of the OLE, the straighter movements in this closed-loop situation may have been due to online monitoring of the decoded trajectory[20] ("re-aiming") and/or to learning processes which optimize directional encoding[18,21]. However, even with this OLE decoder, there was prominent drifting before movement onset, resulting in skewed movement trajectories (Fig. 8a; Supplementary Movie 1). In contrast, trajectories were straighter when using Direct Regression and ANN (Fig. 8c, e; Supplementary Movies 2,3).

To assist calibration of the ANN decoder, its input was seeded with empirically-derived tuning functions (hybrid-ANN, hANN; see Methods). The offset speed encoding model (obtained from the assisted calibration part of the session) was fit to each unit used in the decoder and assigned to a virtual unit in the input layer of ANN. A wide range of velocities was then simulated and used to generate firing rates for each input layer unit following its offset encoding model. These velocities and firing rates were used to train the network. With this hANN decoder, the monkey controlled movement speed well, in terms of holding the cursor in the center before movement onset and stopping at the target after movement offset (Fig. 8f). When compared with the alternate decoder from the same day, the mean cursor motion rate (speed) during the center-hold period was smaller for both Direct Regression and hANN than OLE, and for hANN compared to Direct Regression (Fig. 9a; single-sided Mann–Whitney $U$-test, $p < 0.001$). A similar conclusion holds for the mean rate of motion during the target-hold period (Fig. 9b). Subsequently, the success rate of stopping in the target was higher for Direct Regression and hANN than OLE (Fig. 9d, day 1–4; chi-square test, $p < 0.001$), although the Direct Regression-hANN differences were not significant (Fig. 9d, day 5–6; chi-square test, $p > 0.1$). Cursor trajectories were also straighter when using Direct Regression or hANN compared with OLE, with smaller curvatures (Fig. 9c, day 1-4; single-sided Mann–Whitney $U$-test, $p < 0.01$). The hANN trajectories were slightly straighter than those decoded with Direct Regression (Fig. 9c, day 5–6; single-sided Mann–Whitney $U$-test, $p < 0.05$). These results demonstrate the potential of these new decoders in facilitating skillful BCI control of both direction and speed.

## Discussion

Velocity decoding is now standard practice for extracting BCI control signals from the motor cortex[22,23]. Decoding is usually a two-step procedure in which an encoding model is defined individually for each neuron in a recorded population, followed by an implicit or explicit inversion of this model in which the entire population is considered together to extract the encoded information. As a vector metric, velocity is composed of direction and speed. Direction is a robust, easily decoded movement parameter and acts as a powerful driver to modulate the firing rate of neurons in the motor cortex and many other neural structures[24]. In contrast, accurate decoding of speed has been more elusive[14,25]. A likely reason for the difficulty in decoding speed along with direction is that the two variables interact in a way that precludes linear mathematical inversion of the encoding model. Speed acts as a "gain field" to the directional cosine tuning function. As a result, the influence of speed on firing rate will be greatest for movements taking place along a neuron's preferred-direction axis. If the gain-field was the only speed-related effect on firing rate, speed would be accurately decoded with linear methods. However, the combination of a speed-gain and a speed-offset term, as in the case of Eq. (5), yields an encoding equation that is no longer in a form amenable to standard matrix inversion.

Conventional BCI decoders use encoding models that fail to account for the speed-offset effect and this may cause characteristic distortions of the predicted movement. These distortions fall into two categories. First, it leads to non-zero decoder contributions when the speed should be zero. This causes the BCI output to drift, which could be addressed by directly adding an ad hoc bias or by the subject actively compensating with a counter signal to hold the controlled device still. Second, improper speed decoding results in distortion of the decoded directional signal. The source of this problem is evident in Fig. 4 (right column). In this example, a unit with firing rates generated using the offset

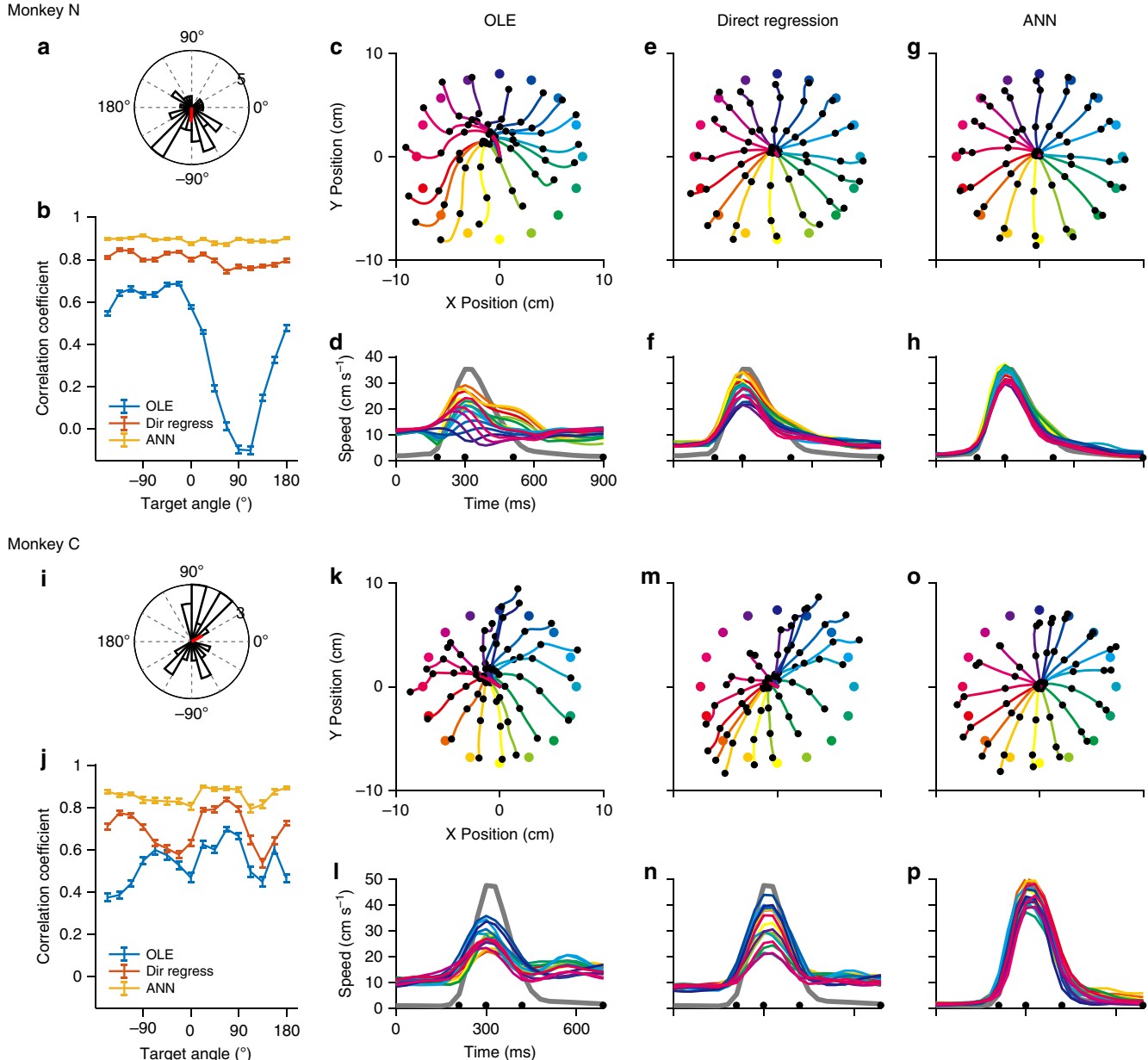

**Fig. 7** Comparisons of performance between minimal OLE, Direct Regression and ANN using arm movement datasets. **a**, **i** Distribution of preferred directions for the selected 33 units for Monkey N and for 19 units of Monkey C. The red line represents the mean of each distribution. **c**, **k** Reconstructed trajectories using minimal OLE for monkeys N and C. **d**, **l** Speed profiles using minimal OLE for each monkey. **e**, **m** Reconstructed trajectories using Direct Regression. **f**, **n** Speed profiles using Direct Regression. **g**, **o** Reconstructed trajectories using the ANN for each monkey. **h**, **p** ANN Speed profiles. **b**, **j** Velocity correlation coefficients between decoded and actual data for all trajectories of monkeys N and C. The correlation coefficients from the ANN decoder were significantly higher than those of the OLE. ($p < 0.001$, $t$-test, Bonferroni). Error bars denote the 95% confidence interval over all trials ($n = 69 \times 10$ trials per target for Monkey N and $47 \times 10$ trials for Monkey C from the 10 repetitions of cross-validation decoding analysis with random data partitions)

model has a non-zero contribution for movements orthogonal to the preferred direction, a movement direction for which units should have a zero contribution. Moreover, instead of having a contribution that mirrors that of the preferred direction movement, the unit is not modulated in the anti-preferred direction. Both speed factors, gain and offset, in Eq. (5) will generate temporal profiles that match speed. However, the combination of contributions enhances the generated profile in the preferred direction and diminishes it in the anti-preferred direction, which can lead to a misestimation of the baseline firing rate and distortions of decoded output. Distorted decoder output is most

evident for population samples showing non-uniform preferred directions (Fig. 5m–p). With a balanced set of preferred directions, the variance due to speed offsets across neurons can cancel out (Fig. 5i–l).

The detrimental effect of offset terms on decoder performance can be illustrated by analyzing the decoded velocities in simulations when the generative model is known (Supplementary Table 1). When the gain-only model is used to generate firing rates, the OLE-decoded velocity components $V_x$ (Fig. 5g) and $V_y$ (Fig. 5h) are accurate, and the population-derived constant, $b_{0P}$, is small (Supplementary Table 1). However, when non-uniform

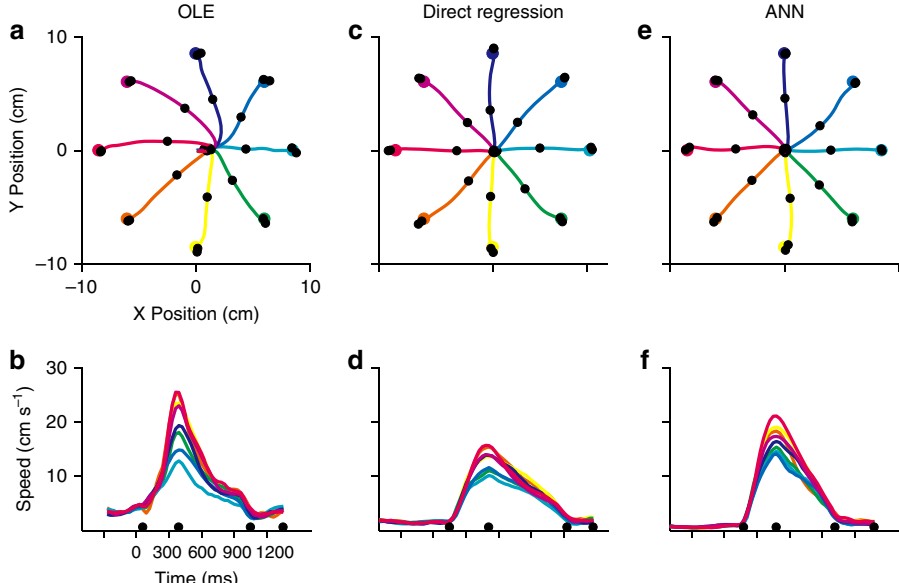

**Fig. 8** Comparison of BCI performance using variance-only OLE, Direct Regression and hybrid ANN decoders. **a**, **c**, **e** Average trajectories in the last sessions (45, 38 and 45 trials per target, respectively) for OLE (day 4), Direct Regression (day 6) and hANN (day 6). **b**, **d**, **f**. Speed profiles for OLE, Direct Regression and hANN. Note that speed profiles were temporally aligned and resampled for presentation, but their magnitudes were not scaled. Time zero represents target onset. Center-hold phase was shortened for OLE to increase success rate and maintain motivation

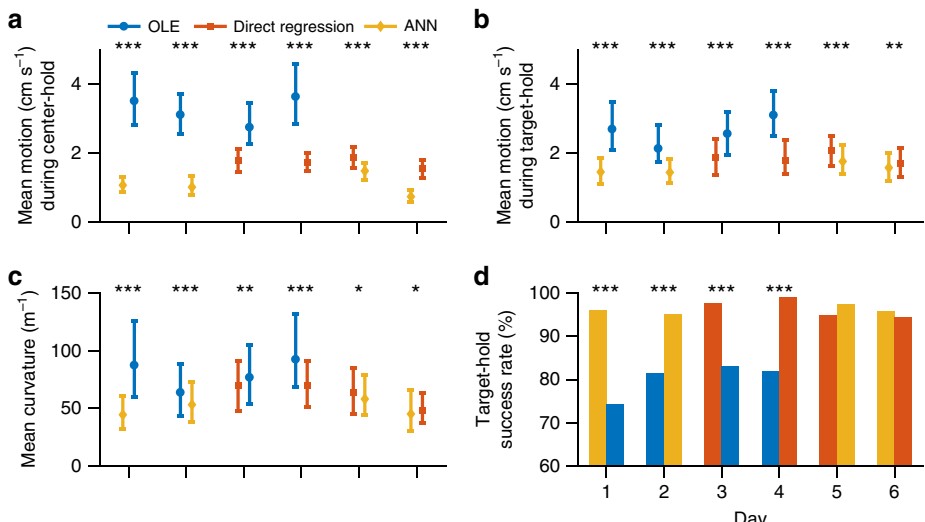

**Fig. 9** Comparison of BCI performance between within-day decoder pairs. For each day, the decoder that was run first is on the left, and the second on the right. Asterisks represent significance level of statistical tests between decoders: *$p < 0.05$, **$p < 0.01$, ***$p < 0.001$ (single-sided Mann–Whitney $U$-test for **a**, **b** and **c**; chi-square test for **d**). On day 2–4, center-hold duration was shortened to be between 200 and 300 ms for OLE. The reference speed $s_0$ used for decoder calibration (see Methods) was lowered to 80% for OLE on day 2–3. Task parameters of each session, including the number of trials per target, could be found in Supplementary Table 2. **a** Mean rate of motion during center-hold before movement onset. The marker (blue circle for OLE, orange square for Direct Regression, and yellow diamond for hANN) represents the median value across all trials, and "whiskers" represent the 25 and 75 percentile values (same in **b** and **c**). **b** Mean rate of motion during target-hold after movement offset. **c** Mean curvature of cursor trajectory between movement onset and movement offset. **d** Percentage of successful target-hold periods of the trials in which the target was acquired

preferred directions are applied to the offset model, the resulting OLE-decoded velocity components $V_x$ (Fig. 5o) and $V_y$ (Fig. 5p) have relatively large population offsets, $b_{sP}$ and $b_{0P}$ (Supplementary Table 1). This suggests that using the wrong decoding model can lead to offset terms across contributing units that do not cancel out. In contrast, for a generative offset model with non-uniform preferred directions, the direct regression-decoded velocity components, $V_x$ and $V_y$, have relatively small population offsets (Supplementary Table 1). This decoder places higher

weights on the units, that when combined with others in the sample, give the best fit to the specified velocity data and this may effectively balance out the offsets across the population. Encoding models that do not account for the speed offset must partition this element of variability into the noise term. In this case, a common source of variance is being added to the noise term and this will induce correlation across neurons. A decoder, such as the "full OLE"[18], accounts for this type of shared variability by normalizing each unit's contribution by its covariance with other

units in the sample and it would be expected that this type of algorithm would handle the offset-generated errors better than those that do not account for noise covariance. However, estimating the covariance matrix between the recorded signals requires a large sample and this may be a barrier in practical BCI use.

A general solution to decoding in the presence of complex encoding may be to use the hybrid approach of combining explicit tuning functions for individual neurons with ANNs[26,27] capable of finding weighting combinations that yield accurate decoded signals. By assigning an empirical tuning function to each element of the input layer in the ANN, a large set of directions and speeds can be used to generate simulated firing rates in this layer to train the network. This can be carried out efficiently and, once trained, the network can work in real time to produce accurate output as it receives the actual firing rates from the recorded units. These networks have proven useful in generating coherent output from nonlinear, complex signals.

Careful examination of movement parameter encoding in simulations and "offline" actual movement data is useful for establishing models and testing putative decoding algorithms. However, actual testing of decoder performance in a BCI context is critical because different control strategies are likely used, leading to unforeseen changes in performance. When directional tuning functions are calculated as monkeys perform center-out arm movements and then recalculated as the same task is carried out in virtual reality using a BCI, the preferred directions of most neurons change[13,21]. There may be multiple reasons for the changes in tuning between the BCI and arm-movement conditions. One set of explanations is based on considerations of the physical plant. A lack of somatic sensory input[28] and/or the non-activation of muscles during BCI[29] may be responsible for changes in the tuning function. Alternatively, open-closed loop changes in the tuning function may be due to short-term and long-term learning that takes place in the BCI paradigm[30–32] as the subject overcomes observed errors in the decoded movement. We tested pairs of decoders in daily experimental sessions with a monkey performing a brain-controlled center-out task (Figs. 8, 9). The paradigm was designed to enforce the control of speed, by requiring the subject to move rapidly and to stop accurately within a target zone. The cursor tended to drift when the OLE decoder was used (Supplementary Movie 1). It took the monkey extra effort, perhaps re-aiming in the direction opposite to the drift, to hold the cursor in the target. The Direct Regression decoder was less prone to this issue (Supplementary Movie 2). When the hANN decoder was used, holding still and stopping at the target was readily achieved in conjunction with rapid movement between the center and peripheral targets (Supplementary Movie 3). When either of the new decoders was used, movements were fast and accurate, with bell-shaped velocity profiles similar to those of actual arm movement.

A basic pillar of neural engineering has been the development of population-based decoders of neural activity. These decoders are often based on engineering principles in which there may be little understanding of the sources generating the signals acting as input to the decoder. This is especially true of decoders that operate as classifiers in which the goal is to detect discrete categories of the parameters contained in extracortical signals recoded with techniques such as EEG, ECOG, and fMRI. A separate class of decoders, derived from empirical observation, describes how parameters are encoded directly in single-unit activity. These model-based algorithms are used in brain-controlled interfaces to extract detailed movement-related parameters from populations of simultaneously recorded action potential waveforms. In some approaches, intended movement is predicted from a combination

of an encoding model and an assumed set of physical characteristics of the motor control system (e.g., smooth trajectories). Nonetheless, almost all intracortical BCIs, including state-space models like the Kalman filter[33,34], operate implicitly or explicitly to invert the encoding models. As we better define the signals present in a recorded sample of neural activity, we are finding that the resulting encoding models are more complex and difficult to invert. This type of problem can be bypassed with direct decoding methods, such as Direct Regression, that do not rely on an initial encoding step. This simple, linear method worked well for the tasks we used in this paper, but cannot account for multivariate interaction between parameters[35]. The inability to account for these dependencies between movement parameters and firing rates decreases the amount of information that could otherwise be extracted. This becomes more important as the number of parameters increases[36]. Another reason to use encoding knowledge in the decoding process is to take advantage of the intrinsic learning that takes place as subjects develop proficiency in BCI tasks[30]. Since this type of plasticity operates on the encoding level, the incorporation of dynamic encoding models can be beneficial. In this regard, hybrid decoders that combine detailed encoding models with nonlinear computational approaches are likely to provide substantial gains in BCI performance. As an initial step, we have shown that our hybrid ANN decoder mitigates the speed-induced errors that persist with conventional decoding schemes. Empirically-derived descriptions of the interactions between behavioral variables and single-neuron discharge, combined with efficient methods of separating and identifying the complex interactions of these parameters in a neural population, will greatly enhance the performance of brain-controlled interfaces.

## Methods

**Behavioral tasks**. Two male rhesus macaques (monkeys N and C) performed 2D center-out reaching tasks on a virtual reality setup. The center-out experimental paradigm has been described previously[37]. A monkey sits in a chair with one hand restrained while the movement of the other hand is tracked using an infrared marker placed on the wrist (Optotrak, Northern Digital). The position of the moving hand is sampled at 60 Hz and projected as a spherical cursor on a 3D monitor (Dimension Technologies). The projected cursor is the only visual feedback of the moving arm. This task consists of 16 targets (radius of 6 mm for Monkey N, 10 mm for C) radially located on a vertical plane in front of the monkey (target distance from the center was 8 cm for Monkey N and 7.4 cm for Monkey C). At the start of each trial, the monkey is required to hold at the center for 200–300 ms, then the center target is extinguished, followed by the selection and display of a peripheral target. The monkey then reaches toward and remains in the target (200–400 ms) to receive a liquid reward. We collected multiple trials to each target in single-day recording sessions: 69 for Monkey N, and 47 for Monkey C. The monkeys made fairly straight movements to all 16 targets.

Monkey N also performed the 2D center-out task using a BCI. The monkey sat in a chair with both hands restrained. At the start of each trial, the cursor was reset to the center. After that, cursor movement in the virtual reality display was driven by recorded neural activity. This task consists of 8 targets (radius of 6 mm) uniformly located along a virtual circle (radius of 8.5 cm) in front of the monkey. The total time the animal had to move the cursor to the target was typically 1500 ms. Both center-hold, before target presentation, and target-hold, after target acquisition, had durations of 400–500 ms.

We compared decoder performance (OLE, Direct Regression, and hANN; described below) by running two sessions of the experiment with different decoders each day. We switched the order of decoders so each decoder was used twice in the first session and twice in the second session of a day across a period of six days. We collected multiple trials to each target across days: 154 for OLE, 181 for Direct Regression, and 183 for hANN. Before these BCI experiments, the monkey had been trained to proficiency on a BCI center-out task with a brief target-hold phase (200 ms) but no center-hold, using the PVA decoder (described below).

**Neural recording**. Monkeys N and C were each implanted with one multi-electrode array (96 channels, Blackrock Microsystems) approximately in the arm area of primary motor cortex (M1). All procedures were in accordance with the guidelines of the US National Institutes of Health and were approved by the Institutional Animal Care and Use Committee of the University of Pittsburgh. We recorded single-neuron responses for arm movement experiments: 86 single units

were isolated for Monkey N, and 93 units were sorted for Monkey C, using the online box-sorting method in the Sort Client program (Plexon Inc.).

For Monkey N's BCI experiments, recorded action potentials were sorted online using box-sorting. On average, the activity of 138 units was recorded in each session, including both single-unit and multi-unit activities (75 and 63 units, respectively).

**Data preprocessing**. Center-out reaches have a stereotypic bell-shaped speed profile. Based on the speed profile of each trial, we normalized time according to standard landmarks. The time landmarks we used are: target presentation, movement onset, maximum speed, movement offset, and hold off. Movement onset and offset were defined as the times when speed reached 20% of the maximum. The time between landmarks was normalized according to the average duration of that epoch across all trials. We implemented this by fixing the number of time bins in each epoch so that the average bin width was 30 ms. Thus, corresponding epochs across trials had the same number of bins. For Monkey N, this consisted of 31 bins divided into epochs of 7, 4, 7, and 13 bins, with average bin widths ($\pm$SE, $n = 69 \times 16$ trials) of $29 \pm 0.11$ ms, $32 \pm 0.21$ ms, $32 \pm 0.25$ ms and 29 $\pm 0.17$ ms respectively. For Monkey C, the task was divided into 8, 3, 4 and 9 bins with average bin widths ($\pm$SE, $n = 47 \times 16$ trials) of $29 \pm 0.07$ ms, $31 \pm 0.12$ ms, 32 $\pm 0.22$ ms, $31 \pm 0.25$ ms. Spikes were counted within each bin and divided by the bin width to estimate firing rate. Position was spline-interpolated and then resampled to match the number of bins in each epoch. Velocity was then recalculated using resampled position. The spike rates were smoothed with a Gaussian kernel (50 ms SD). The position and velocity data recorded in Monkey N's BCI sessions were similarly aligned at those landmarks. However, velocities were not adjusted by bin widths to preserve their original magnitudes (Fig. 8b, d, f).

**Simulation**. Two sets of simulations were performed to show how speed might affect firing rate. One set consisted of the parameters $m = 0.5$ Hz per m s$^{-1}$, $b_s = 0$ Hz per m s$^{-1}$, $b_0 = 30$ Hz and $\tau = 0$ ms. Here we use the subscript $i$, to refer to the $i^{\text{th}}$ unit in a sample of N:

$$y_i(t - \tau) = 30 + 0.5|\mathbf{v}(t)|\cos(\theta(t) - \theta_{\text{PD}i}) + \varepsilon_i(t) \tag{8}$$

and the other was $m = 0.25$ Hz per m s$^{-1}$, $b_s = 0.25$ Hz per m s$^{-1}$, $b_0 = 30$ Hz and $\tau = 0$ ms:

$$y_i(t - \tau) = 30 + 0.25|\mathbf{v}(t)|\cos(\theta(t) - \theta_{\text{PD}i}) + 0.25|\mathbf{v}(t)| + \varepsilon_i(t) \tag{9}$$

Equation (8) is an example of the 'gain-only' model and Eq. (9) corresponds to the 'offset' model.

The preferred direction angle is a free variable in Eqs. (8) and (9). Preferred directions for $N = 36$ simulated neurons, $\theta_{\text{PD}i}$, were chosen from either a uniform or a non-uniform distribution. The PDs of the uniform distribution were spaced at 10° intervals. The non-uniform distribution was chosen to be the von Mises distribution: $\theta_{\text{PD}} \sim \text{von Mises } (\mu, \kappa)$, where $\mu$ is the angle of central tendency and $\kappa$ is the concentration parameter[38]. We chose $\mu$ as 180° with $\kappa = 1.3$. The simulation was based on the behavior of Monkey N. The simulated dataset consisted of 800 trials (50 repetitions for each of 16 targets, see Fig. 1) with a single speed profile (|**v** (t)|, the mean profile of all $69 \times 16$ trials from Monkey N). Each trial had 31 bins (bin width = 30 ms). Simulated movements to each of the 16 targets were straight. Poisson realizations of the underlying rate parameter from Eqs. (8) or (9) served to generate binned spike counts which were divided by the bin width and smoothed the same way as real data, to convert to simulated firing rates. These rates were used for encoding velocities and subsequently, for testing the performance of decoders in predicting trajectories. Ten-fold cross validation was used for decoder performance evaluation, and the procedure was repeated ten times with randomly generated data partitions.

**Decoders**. Whereas *encoding* models predict a unit's firing rate using movement parameters (in this case direction and speed), *decoding* algorithms use sampled firing rates to predict movement parameters. Most intracortical decoding algorithms mathematically invert these models to predict movement parameters and we analyzed the performance of one such decoder, the OLE, using a variety of simulated and actual data. We then tested alternative decoders that do not rely on encoding inversion. Decoder performance was evaluated with arm movement data using 10-fold cross validation (repeated ten times with random data partitions).

**Population vector algorithm**. The direction-only encoding model (Eq. 2) is used to find the $i^{\text{th}}$ cell's modulation depth, $m_i$, the unit-length preferred direction in vector form $\mathbf{pd}_i = [b_{xi}, b_{yi}]/m_i$, and its offset constant, $b_{0i}$, for use in the population vector algorithm. Each unit in the recorded sample contributes to the population vector in the form of a vector $\mathbf{c}_i(t) = r_i(t)\mathbf{pd}_i$ which lies in the unit's preferred

direction with a magnitude equal to its normalized instantaneous firing rate:

$$r_i(t) = \frac{y_i(t) - b_{0i}}{m_i} \tag{10}$$

The vectorial contributions from all included units are added to yield the population vector $\mathbf{pv}(t)$:

$$\mathbf{pv}(t) = \sum_{i=1}^{n} r_i(t)\mathbf{pd}_i \tag{11}$$

The decoded velocity $\mathbf{v}_{\text{pred}}(t)$ is computed as:

$$\mathbf{v}_{\text{pred}}(t) = \frac{k_s}{n}\mathbf{pv}(t)$$

where $k_s$ is a speed factor that converts the magnitude of the population vector to a physical speed, and $n$ is the number of units used for decoding[18]. Note that although this algorithm has no terms accounting for speed, there is an emergent correspondence between the population vector magnitude and speed[3]. This is because speed acts as a gain factor on the directional tuning curve, as seen for example in the gain-only encoding model (Eq. 6). Also note that although its performance is not analyzed here, PVA demonstrates how inversion of the encoding model is used for decoding (e.g., Fig. 4), and it is closely related to the OLE algorithm explained next.

**Optimal linear estimator**. The OLE[39] can be considered a modification of the population vector algorithm. Both rely on previously estimated preferred directions and normalized firing rates. The OLE operates by inverting the encoding equation

$$\mathbf{r}(t) = \mathbf{B}\mathbf{d}(t) + \varepsilon(t), \tag{12}$$

where $\mathbf{r}(t)$ is a $n \times 1$ vector consisting of the normalized firing rates for each sampled unit (Eq. 10) for a movement in direction $\mathbf{d}(t)$ ($2 \times 1$). Since the movements considered here are two-dimensional, $\mathbf{B}$ is the $n \times 2$ matrix of unit-length preferred directions ($\mathbf{pd}_i$). Thus Eq. (12) is just a reformulation of the decoding parameters found in the direction-only model of PVA. The key aspect is that OLE treats the matrix of preferred directions $\mathbf{B}$ as the explanatory variable in an ordinary multiple regression model. Hence, the least squares estimator for direction is:

$$\mathbf{d}_{\text{pred}}(t) = (\mathbf{B}'\mathbf{B})^{-1}\mathbf{B}'\mathbf{r}(t). \tag{13}$$

For ease of notation, Eq. (13) can be written as $\mathbf{d}_{\text{pred}}(t) = \mathbf{P}\mathbf{r}(t)$. The matrix $\mathbf{P} = \alpha(\mathbf{B}'\mathbf{B})^{-1}\mathbf{B}'$, with dimensions $2 \times n$, is analogous to an optimized version of the matrix $\mathbf{B}'$. The scaling constant, $\alpha$, is chosen such that the average length of the vectors in $\mathbf{P}$ is unit length. Once $\mathbf{P}$ has been calculated, it is then used to calculate population vectors and trajectories (Eq. 13). Note that if the sample of preferred directions, $\mathbf{B}$, is distributed uniformly about the unit circle, $\mathbf{B}'\mathbf{B} = \mathbf{I}$, where $\mathbf{I}$ is the identity matrix, and $\mathbf{d}_{\text{pred}}(t) = \mathbf{B}'\mathbf{r}(t)$ which is the definition of the PVA (Eq. 11). Thus, for a uniform sample of preferred directions, PVA and OLE are equivalent and both are optimal[40].

Because the PVA and OLE rely on previously estimated direction-only encoding parameters, their performance can actually be degraded by extensions that account for speed. Consider a modified form of the offset encoding model (Eq. 4):

$$y_i(t - \tau) - b_{0i} = |\mathbf{v}(t)|\left(b_{xi}d_x(t) + b_{yi}d_y(t) + b_{si}\right) + \varepsilon_i(t)$$

Since speed is itself one of the parameters we wish to estimate, this equation cannot be inverted in the same way as the direction-only model (Eq. 2). The least squares solution described in Eq. (13), technically the "minimal OLE"[18], is valid only when the noise terms of all units are independent, identically distributed random variables. If there is correlation between the noise terms of different units, the OLE decoding preferred direction matrix should be

$$\mathbf{P} = \alpha(\mathbf{B}'\boldsymbol{\Sigma}^{-1}\mathbf{B})^{-1}\mathbf{B}'\boldsymbol{\Sigma}^{-1}, \tag{14}$$

where $\boldsymbol{\Sigma}$ is the covariance matrix of noise terms. Implementing this "full OLE" requires estimating the full covariance matrix, which is challenging when the dimensionality is high and the data are limited[18]. The "variance-only OLE" is often used for online closed-loop control, where only the diagonal of the covariance matrix is used to find the decoding preferred directions. Instead of assuming that the variance terms for each unit are the same (minimal OLE), the variance-only OLE uses each unit's variance.

**Direct regression**. Instead of defining encoding parameters for each unit as a precursor to decoding, we can estimate the least squares vector weights of individual neurons jointly from the collective ensemble of sampled firing rates. This procedure is referred to as "Reverse or Direct" Regression[40] and obviates the problem of inverting the parameter matrix for speed coding.

Using notation corresponding to our previous descriptions:

$$\mathbf{V} = \mathbf{YB} + \varepsilon,$$

where $\mathbf{V} = [\mathbf{v_x}, \mathbf{v_y}]$ ($T \times 2$) is a matrix of $t = 1,\ldots,T$ sampled velocities from center-out trials, $\mathbf{Y} = [\mathbf{y_1}, \mathbf{y_2},\ldots,\mathbf{y_T}]'$ is a $T \times n$ matrix of firing rates (each of the $n$ neurons is an explanatory variable) and $\mathbf{B} = [\mathbf{b_x}, \mathbf{b_y}]$ is an $n \times 2$ matrix of regression coefficients. The least-squared solution for $\mathbf{B}$ is:

$$\mathbf{B} = (\mathbf{Y}'\mathbf{Y})^{-1}\mathbf{Y}'\mathbf{V}$$

Note that we can include an overall population offset term if desired. Since the components of velocity are orthogonal in this center-out task, and the task is balanced with equidistant targets located on a circle, the above formulation is equivalent to separate regressions for $\mathbf{v_x}$ and $\mathbf{v_y}$:

$$\mathbf{v_x} = k_x + \mathbf{Yb_x} + \varepsilon_x$$

$$\mathbf{v_y} = k_y + \mathbf{Yb_y} + \varepsilon_y,$$

where $k_x$, $k_y$ are regression constants common to the whole sample.

The estimator for velocity is then:

$$\hat{v}_x(t) = k_x + \sum_{i=1}^{n} b_{xi} \cdot y_i(t) \qquad (15)$$

$$\hat{v}_y(t) = k_y + \sum_{i=1}^{n} b_{yi} \cdot y_i(t) \qquad (16)$$

**Artificial neural network**. The PVA, OLE and Direct Regression decoders all adopt some linear mapping to predict velocity from population firing rates. These decoders could be suboptimal given that neuronal firing rate may be better described by some nonlinear model, such as the offset model of Eq. (4). ANNs are known as good approximators for nonlinear functions and could be used to find a nonlinear mapping from firing rates (input) to velocities (output).

Here we consider a single-hidden-layer feedforward network as the ANN decoder. The value of the $j^{th}$ hidden-layer neuron in the ANN is

$$h_j(t) = \tanh\left(b_j + \sum_{i=1}^{n} w_{ij}y_i(t)\right),$$ where $y_i$ is the value of the $i^{th}$ input-layer

neuron (i.e. firing rate of the $i^{th}$ recorded unit), $n$ the number of input-layer neurons, $w_{ij}$ the weight parameter between the input layer and the hidden layer, $b_j$ the bias, and tanh the hyperbolic tangent transfer function. The output of the ANN is predicted velocity, given by

$$\hat{v}_x(t) = b_x + \sum_{j=1}^{m} w_{xj}h_j(t) \qquad (17)$$

$$\hat{v}_y(t) = b_y + \sum_{j=1}^{m} w_{yj}h_j(t), \qquad (18)$$

where $w_{xj}$ and $w_{yj}$ are weight parameters between the hidden layer and the output layer, $m = 10$ is the number of hidden neurons, and $b_x$ and $b_y$ are the biases. Parameters $w_{ij}$, $w_{xj}$, $w_{yj}$, $b_j$, $b_x$, and $b_y$ are randomly initialized, and then learned from training data using the backpropagation algorithm, with the goal of minimizing the mean squared error between predicted and ground truth output. Early stopping was used to prevent overfitting. This was achieved by dividing data into a training and a validation dataset. Training data were used to iteratively adjust network parameters using backpropagation, and validation data were used to monitor network performance (the mean squared error in the output) and stop training when performance stopped improving. To decode the simulation and arm movement data, two trials from each target were used for validation. For online BCI control, we generated artificial data (described below) to train the network and recorded real data were used for validation. We implemented this hANN decoder using the Matlab Neural Network Toolbox (version 8.4; Mathworks Inc.).

**Decoder calibration for closed-loop control**. For BCI experiments, a decoder was calibrated at the beginning of a session using the assisted control paradigm[6]. During the calibration procedure, cursor movement toward the target was 'assisted' by combining the decoded movement intention with an automated idealized command:

$$\mathbf{v}_{ctrl}(t) = \alpha\mathbf{v}_{auto}(t) + (1 - \alpha)\mathbf{v}_{pred}(t), 0 \leq \alpha \leq 1$$

where $\mathbf{v}_{auto}$ is the automated velocity command, $\mathbf{v}_{pred}$ the predicted command by the decoder, $\mathbf{v}_{ctrl}$ the actual control signal driving the cursor, and $\alpha$ the amount of assistance. The direction of $\mathbf{v}_{auto}$ was always pointing to the target from the current position. The magnitude of $\mathbf{v}_{auto}$ was

$$|\mathbf{v}_{auto}(t)| = 4\beta_{inc}(t)\beta_{dec}(t)s_0$$

$$\beta_{inc}(t) = \min\left(1, \frac{t}{d(t)/s_0}\right)$$

$$\beta_{dec}(t) = \min\left(1, \frac{d(t)}{d_0}\right),$$

where $t \geq 0$ is the time since target onset, $d$ is the distance from current position to target, $d_0 = 8.5$ cm is the distance from starting position to target, and $s_0 = 15$ cm s$^{-1}$ is a reference speed. This formula generates a bell-shaped speed profile for $\mathbf{v}_{auto}$. The decoder was re-calibrated using $\mathbf{v}_{ctrl}$ and recorded neural firing rates in successive blocks of movements to each target. The amount of assistance, $\alpha$, started with a value of 1 in the first block of trials, and was decreased by 0.2 after each subsequent block. The last re-calibration was done after completion of the sixth block in which assistance had decreased to 0. The latest decoder was then used for the rest of the session, giving the monkey full control of cursor movement. The gain-only regression model was used to select units with $R^2 > 0.03$ for any decoder. This ended up with 62, 57, and 58 units (of which 41, 39, and 39 were single units and the rest contained multi-unit activities) on average for the OLE, Direct Regression and ANN decoder, respectively.

The hybrid-ANN (hANN) decoder calibration was initiated by assigning the empirical offset-tuning function (Eq. 4, found using data from previous assisted calibration blocks) for each included recorded unit $i$ to an input of the ANN. A wide range of velocities (in 16 uniformly arranged directions, including eight new directions in addition to the 8 target directions) was generated using the automated control signal ($\mathbf{v}_{auto}$). These velocity profiles were then fed into the tuning function for each input unit $i$ to produce firing rate profiles, $y_i$. Firing-rate data of each input unit $i$ in five trials for each of the 16 directions were generated with Poisson noise using $y_i$ as the rate parameter. These artificial firing rates and velocities were used to train the network. Real firing rate and velocity data recorded from the assisted calibration procedure were used to validate the network. Early stopping was adopted to avoid overfitting and training was terminated when performance on the real data was no longer improving. In practice, it took about 9 s to train the randomly initialized hANN decoder with 10 hidden-layer units in a computer with a 2.8 GHz CPU and 16 GB of memory. Making a prediction of intended velocity from population firing rates took about 8 ms.

**Code availability**. Code package for offline data analyses is available from the corresponding author on reasonable request.

## Data availability
The data sets recorded and simulated for this study are available from the corresponding author on reasonable request.

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

## Acknowledgements

The authors wish to thank Dr. Andrew Whitford and Xiao Zhou for collecting neuro-physiological and behavioral data of arm movement experiments. We thank Carmen Fernandez Fisac for assistance in BCI experiments. Dr. Steven Chase gave valuable advice in the use of the OLE decoders. We would also like to thank Dr. Elizabeth Tyler-Kabera for her help with the surgeries.

## Author contributions

Y.I. conceived the study, with contributions from A.B.S and S.B.S. Y.I., J.O., H.M., and S.B.S. analyzed simulation data and arm movement data. H.M. and A.B.S. designed the hANN decoder and the BCI experiments. H.M. recorded and analyzed BCI data. Y.I., A.B.S., J.O., and H.M. wrote the manuscript. S.B.S. provided feedback on the manuscript.

## Additional information

**Competing interests:** The authors declare no competing interests.

