## [Peer Review File · Nature Communications]

Reviewers' Comments:

Reviewer #2:

Remarks to the Author:

Inoue and colleagues present experimental and computational results to highlight issues with decoding velocity from activity of neurons in motor cortex. Specifically, they emphasize the distinction between direction of movement and speed, which both have been shown to contribute to motor cortex activity. They show that decoders designed based on encoding models that do not incorporate speed can produce distortions, and that a decoding schemes without explicit encoding models ('direct regression') can avoid these pitfalls. The arguments are overall well-supported with thorough simulation and analytical treatments of the problem, and supplemented by offline decoding analyses on experimental data. The problem is also certainly of interest for brain-machine interfaces, since stopping/holding is a well-documented issue with many of these devices. Unfortunately, the manuscript as presented has a rather narrow scope/focus that limits its impact, and may therefore be better suited to a specialized journal.

In sum, the manuscript presents a few findings: 1) motor cortex firing contains information about both speed and direction with both a gain-factor and offset, 2) decoding algorithms (PVA and OLE) that use a direction-only encoding model produce skewed trajectories, 3) direct regression does not suffer this issue. Point 1 has already been made by this group, so there is no new substantial novel findings on neural encoding presented. Points 2 and 3 are thoroughly demonstrated with both simulated data where neurons have known encoding schemes and with offline decoding of experimental data, as well as some analytical treatments to give further grounding. This thoroughness is arguably a strength of the manuscript. But ultimately all of these treatments show the same thing but in slightly different ways. That is, they don't provide substantial insights, but just further reiterate these points almost to a repetitive degree. I think the authors' larger point—that decoders without strong priors on encoding models may function better—is a very useful argument of broad interest to both brain-machine interface and neuroscience communities and I wish the work were more centered around this line of argument. But the current manuscript presents a hyper-focused analysis on one specific example of encoding/decoding that loses this thread a bit. Sharpening the presentation, and perhaps providing evidence for other examples of encoding/decoding problems where direct regression could help, would be one strategy for improving the impact. Alternately, if the authors wish to focus on the decoding speed for BMI applications, more work with closed-loop control demonstrations that Direct Regression leads to improved performance would be a strong way to increase impact.

Specific points/comments:

-Does the number of analyzed units represent unique samples of neurons (given that these recordings are from a chronically implanted fixed array)?

-I found the mathematical presentation somewhat difficult to follow because the same variables are reused repeatedly across different models. While I appreciate that this is done to maintain consistent meaning for a given variable across related models, it does increase the burden on the reader to keep track of what model is being discussed.

-Given that the authors have access to experimental data, I found it a bit strange that they present both simulations with—basically—arbitrarily chosen parameters and simulations with parameters derived to match the experimental data. The single neuron examples do help provide some intuition (e.g. fig 4). But from an argumentation perspective, I am unclear what is the benefit of figure 5 (and tables 1 and 2) versus fig 6 is. I understand that fig 5 data allows the authors to manipulate the properties of the ensemble (e.g. direction tuning distributions). But the authors do not seem to use this ability to present a specific finding/point to build a bigger argument. Either the ability to manipulate population properties in simulation needs to be more fully exploited (i.e. also manipulating other properties of the population like offset ratio distributions, etc) to highlight some key point(s)

relevant to the manuscripts overall claims/argument, or the presentation needs to be streamlined.

-The comparison of OLE to direct regression in fig. 7 is a bit unfair/misleading given the substantial difference in the total number of units contributing to each. A matched population decoding comparison must be provided.

-“Interestingly, when the excitability...” paragraph should also mention that recent results from Perel et al. (J Neurophysiol 2015) suggest threshold-crossings and multi-unit data seem to contain significant speed information.

Reviewer #3:

Remarks to the Author:

A central tenet of system neuroscience is that the firing rate of neurons represent (that is, encode) parameters of sensory, motor, and cognitive processes. Neuroscientists have mostly focused their efforts in finding out how changes of specific parameters affect neural activity. However, the problem to solve for neuroprosthetics is the reverse, that is, how changes of neural activity can be decoded in terms of behavioral parameters. Andy Schwartz and colleagues have made key contributions over the years on both sides of this equation.

In the last few years we have seen great advances in neural prosthetics of the arm and hand. However, the decoding of movement speed has remained more problematic than the decoding of movement direction, which is a critical problem for the fine control of prosthetic arms. The authors show that this problem arises from how motor cortical neurons encode movement direction and speed, and from some of the methods used for decoding neural activity. Specifically, the spiking activity of motor cortical neurons changes with movement direction and speed. However, the activity of many neurons is modulated by both the velocity vector (direction and speed), and by an additional speed-related offset (b_s) that is direction independent. The meaning of this speed-related offset has received little attention so far.

Here the authors investigated the effect of this offset on the decoding performance of popular decoders. In particular, they show that the speed-related offset creates a ‘distortion’ in the movement trajectory decoded by some extraction methods. This is because, the contribution of neurons with a preferred direction at 90 deg from the movement direction should be neutral (i.e., $\cos(90 \text{ deg})=0$) compared to that of neurons with other preferred directions. However, the presence of a speed-related offset means that the activity of these neurons is not neutral (see Fig. 4B, offset model) when a movement is orthogonal to their preferred direction, and that in a population vector decoding scheme they contribute to ‘pull’ the decoded trajectory in the wrong direction.

The article is clearly written and provides important new insight in motor cortical movement velocity coding and decoding that is of broad interest to both neuroscientists and neural engineers.

My only main critical comment is in regard to the distortion effect of the constant b_0 before and after the movement for the offset model which is illustrated in Fig. 4 C, D, and E, as well as in Fig. 5N, Fig. 6C, Fig. 7C, and I. The mis-estimated baseline (b_0) (p. 9, line 302) is the result of using movement-related activity in the regression function. Since the change of activity of many neurons as a function of direction is not symmetrical around their baseline activity (Fig. 3B and C, in contrast to 3A), the offset (b_0) found using movement-related activity will be necessarily different than their actual baseline. A simple way to deal with this is to compute the regression using neuronal activity minus the recorded baseline. So, for example Eq. 4 would become:

$$y_i(t) - b_{0_i} = m_i |V(t)| \cos(\theta - \theta_{PDi}) + b_{si} |V(t)| + \varepsilon_i$$

with b_{0_i} being the activity recorded during the baseline period (rather than a regression parameter).

In this case, the baseline would necessarily correspond to the correct baseline and reduce the distortion of decoded kinematics before and after the movement period. However, the distortion in the decoded movement trajectory, as seen in some conditions, would remain essentially the same as illustrated in the article since that distortion has a different cause (as mentioned above).

Giuseppe Pellizzer

Minor points

p. 2. The text (line 65) indicates that the average trajectories plotted in Fig. 1A were for Monkeys N and C, whereas the legend of the figure indicates that they were only for Monkey N. Please reconcile.

p. 3. Line 74. "We recorded single neuron responses: 86 for Monkey N and 93 for Monkey C." please indicate how single neurons were isolated. (Incidentally, it would be interesting to know how important for decoding is it to isolate activity of single neurons relative to using multiunit activity. This is not the object of this study, but, if the authors find it appropriate to include it in the text, it would be interesting to know what their position on this is.).

p. 3. Lines 93-95. I understand the need for resampling, but I am not sure what precisely is meant by "to match each bin with the velocity values adjusted by bin width".

p. 4. Since a time-dependent direction of movement $D(t)$ was part of Eq. 2, then its vector formulation in Eq. 3 should also have a time-dependent variable, that is, $\theta(t)$. Same comment for Eq. 5, 7, 9, and 10. However, since movement direction was approximately constant from the center to the target, then t could be omitted in Eq. 2 (and following Eqs.), and mention that movement direction was considered to be straight and t was omitted here for simplicity.

p. 4. Line 120. I am not sure why the velocity magnitude $|V(t)|$ is said to be "proportional to speed" instead of "equal to speed". Please clarify.

p. 8. Line 261. "...poison noise..." should be corrected with "...Poisson noise...".

p. 8. Eqs. 23, 24 The regressions were computed using firing data averaged over a time-window of interest, presumably between movement onset and offset; however, please specify the time window used for the regressions.

p. 13. Line 386. The central tendency (red arrow) of the von Mises fit in Fig. 6A is at negative 147 deg. Please correct the sign in the text.

Fig. 7. The thick gray line that represents average speed in the C, E, I, and K plots should be mentioned in the legend.

Reviewer #4:

Remarks to the Author:

General

The main goal of the paper, as well summarized in the abstract, is to solve a difficulty in extracting speed from velocity from motor cortex neuronal activity. They suggest that this difficulty is due to the way speed is encoded by individual neurons and demonstrate how some encoding-decoding procedures used by standard extraction algorithms can produce characteristic errors. They conclude

that the problem can be reduced by alternative extraction algorithm that bypasses explicit neuronal parameter encoding using the reversed (direct) regression (Kass 2005).

The question posed is interesting and significant. We may still debate about the "the way speed is encoded by individual neurons". It may well be that the brain does not necessarily use the explicit encoding of any parameter that we the observers attempt to decipher. This way or the other, it seems that the paper suggestion to bypasses explicit neuronal parameter encoding may provide a solid solution as for the optimal algorithm for BMI.

In general, this paper could serve researchers in the field especially in new ways to examine the population vector approach in neurophysiology. The paper is well written and the general approach is solid. While It has a potential for clarifying important issues with using the results for advancing brain-machine interfaces, the paper does not address it directly. The paper is based, probably too heavily, on past studies of the group, with data from a previous study and with an algorithm from Kass (2005!). These and other concerns are listed below.

Disclosure: I would like to declare that papering the review, I consulted one of my colleges

Concern 1: Novelty: The Authors propose an alternative extraction algorithm that bypasses explicit parameter encoding", the alternative method is a simple linear regression. The first thing any engineer would try on a new prediction problem; The paper presents the PVA and OLE methods for predicting movement velocity. These methods have become the de-facto standard in physiology texts, explaining how movement is related to fitting rates of neurons in the motor cortex and how this relation can be used to estimate velocity. They suggest that conventional methods that make use of fitted neuronal rate models are biased in the general case of non-uniform neuronal distributions, and when the firing rate is modulated with respect to the magnitude of the velocity. This is, WE believe true but not surprising. Given the popularity of the PVA model (~2000 citations of the 1982 Georgopoulos paper and ~650 citations of the Moran & Schwarz paper) and the numerous computational studies which used this model to predict movements.

The authors could clarify more vividly the contribution of current analyses add to our current understanding by addressing some of the suggestions below.

Yet, as stated above, the notion of using alternative extraction algorithm that bypasses explicit parameter encoding, is attractive and will be hopefully used in new ways to demonstrate novelty and a new understanding.

Concern 2: We wonder if the suggested algorithms represent the optimal approach to bypasses explicit parameter encoding. At this stage, contemporary science novel methods and algorithms are being developed in other regimes of machine learning and advanced controllers. Neuroscience will be advanced by applications of more visionary algorithms and test them against the linear regression approach. Algorithms that could be used to facilitate understanding of the motor cortex may include dynamic-adaptive controllers that can examine how past observations predict the future ones (like the brain) and adapt rapidly (like the brain) to create updated control of actions. Specifically, the paper and its message could become more attractive by modelling of the causal relationships in the brain-behaviour loop, based on the notion of adaptive-predictive features of internal models of the brain. However, the data used here is probably not optimal for this purpose.

Concern 3: The paper proposes to put aside process-based mechanistic models in favor of standard (even non-regularized) linear regression from firing rates to instantaneous velocity (direct regression). The proposed solution (at this stage and version of the paper) is not yet convincing and could be a non-optimal compromise. It is non-mechanistic, providing no new insights on how the motor cortex is involved in motor control and probably too simplistic from the perspective of a BMI engineer (yet to be

tested). Previous studies used of the Kalman filter (Wu and Donoghue) and kernel-based regressive models (KARMA, Shpigelman et al.) have shown how methods that also model movement dynamics and allow for inter-dependency between neuronal firing rates to affect the prediction of movement can out-perform simple linear regression. These and other approaches should be compared in the unbiased and extensive manner to the methods tested here. In sum, a critical comparison of direct-linear regression to other algorithms is insufficient at this stage of the paper. We can only guess that algorithms, which optimize discovery of causal relations between brain activity and behavior, would have to focus on the adaptive, predictive nature of the brain as a controller that dynamically processes, represents and executes the desired actions like movements with desired velocity profiles.

The implication of the discussion (lines 488-502) that algorithms like Kalman-filter, which "rely on fitting encoding models to the firing rates of each unit in the recorded sample", are suffering from problems related to difficulty in inverting the models, is misguided. The difficulty is a specific feature of the PVA and OLE models. However, the direct regression (and Kalman filter or KARMA) do not suffer from this separation of model fitting criterion and velocity estimation error (they are fitted to minimize the error in velocity estimation). The 'problem' with the direct method is that it should not be interpreted, without validation, as providing neuronal tuning (because parameters that refer to one neuron would end up different if they were fitted with or without observation of the other neurons. If the authors think otherwise, we suggest that they try and make their case empirically (for example, by comparing the models on a test set not used for fitting the models (see other concerns, item 1 below)

Other main concerns

The analysis of accuracy is not implementing the standard separation of training (/fitting) from testing (/evaluation) data. This separation allows evaluation results to reflect expected performance on unseen data (as would be the case when models are in use). When the results are shown on data used also for fitting the models, the evaluation reflects how good a fit the models are, which, for richer models would typically be better. Although this study shows that the prediction methods are systematically biased, there is no reason to mix between effects of model complexity on fitting and the evaluation. This is taken care of by simply fitting the models on a training set and giving evaluation results on a test set.

There are several ways of treating the problem of biased decoding by OLE and PVA, which are sub-optimally fitted for their eventual use. One solution, as proposed by the authors, is to abandon the PVA and OLE as presented in this study and opt for a direct linear regression method. As explained above (concern 3), may be a problematic solution.

Another way would be to find a method of fitting the OLE and PVA parameters so that they directly minimize the decoding error. This would assume that the models represent a mechanistic process by which the neuronal activity is causing the end-point velocity (through some presumed kinematics to hand dynamics transformation).

Yet another approach would be to encompass the assumptions of what neurons encode (how their firing rates are affected by movement velocity) into a probabilistic generative model of firing rates and find ways to fit them in the probabilistic modelling framework so that the resulting model can be used to estimate the underlying hidden ("unobserved") velocity variables from the observed firing rates. This, for example, is the approach taken by the Kalman filter model, which assumes linear state dynamics with Gaussian noise (the modeler can decide what to represent by the state E.g. the state can be the current position, velocity and acceleration) and a similar linear-Gaussian observation model.

A probabilistic modelling approach that might produce a better decoding method than the described PVA method while still proposing a relation between intended movement and cortical activity and can

be physiologically motivated.

Finally, other approaches can apply new neural networks (NN) models, with an attempt to feature the local circuits in cortex and predict how neuronal state changes when instructed to choose velocity in a given context. While NN suffered a long era of "depression" in previous years, new advances suggest that biologically-relevant networks can be used to find new ways to study the fundamental problem of how does neuronal population is modulated during control of movements

Minor comments

1. Some of the notation is confusing. Most papers use single characters to represent variables. Non-capitalized bold characters are used for vectors and capitalized characters (typically bold) are only used for matrices
2. We don't think the paper says how the PVA and the OLE parameters are estimated from recorded movements. (WE could have missed it?). We guess they are estimated to minimize the mean squared error per unit but this is not stated. This information is important to understanding the claims of the study.
3. figure 2: were offset ratios calculated for the best fitted time lag? Is the best fit calculated for each time lag and then the best fitting result kept? This is not clear.
4. It is not clear if the authors chose to use stretched/ compresses time bins (so that movements are time aligned) because of their previous experience or did they test the possibility that these variations are relatively small and the movements are relatively uniform in their time/speed profiles). If this could have been avoided, trade-offing simpler descriptions for the price for some complexity in coding the data processing.

We would like to thank the three reviewers for their supportive and detailed comments. We have embraced these remarks, made extensive revisions, developed an additional decoder, and performed an additional set of experiments showing how these new decoders performed during BCI experiments. A common thread through the reviews was to demonstrate how knowledge of speed encoding and the development of decoders that account for this can improve, BCI performance. We have very nice results showing this, now. In addition, we developed a new hybrid decoder that uses empirical speed encoding to initialize an artificial neural network for rapid training. This was one of the decoders used in the BCI work and it worked very well. Finally, we cleaned up and streamlined the manuscript to emphasize the main point and shifted a fair amount of description to the supplementary material.

Specific Reply:

Reviewer #2

Does the number of analyzed units represent unique samples of neurons (given that these recordings are from a chronically implanted fixed array)?:

Data were collected on single-day recording sessions. Each example is from one recording day. Analyses using arm-movement data from each monkey was recorded on a single day. Each combination of paired decoders for the BCI experiments were tested on single days. Text has been added to the manuscript clarifying this point.

I found the mathematical presentation somewhat difficult to follow because the same variables are reused repeatedly across different models. While I appreciate that this is done to maintain consistent meaning for a given variable across related models, it does increase the burden on the reader to keep track of what model is being discussed.

-Given that the authors have access to experimental data, I found it a bit strange that they present both simulations with—basically—arbitrarily chosen parameters and simulations with parameters derived to match the experimental data. The single neuron examples do help provide some intuition (e.g. fig 4). But from an argumentation perspective, I am unclear what is the benefit of figure 5 (and tables 1 and 2) versus fig 6 is. I understand that fig 5 data allows the authors to manipulate the properties of the ensemble (e.g. direction tuning distributions). But the authors do not seem to use this ability to present a specific finding/point to build a bigger argument. Either the ability to manipulate population properties in simulation needs to be more fully exploited (i.e. also manipulating other properties of the population like offset ratio distributions, etc) to highlight some key point(s) relevant to the manuscripts overall claims/argument, or the presentation needs to be streamlined.:

We revised the naming/description of the variables to be more consistent. The analyses are introduced with better descriptions of the rationale and how they fit into the overall theme of the presentation. The original Figure 6 and associated analysis have been removed, as we agree they were unnecessary. These changes should make the reason for the remaining simulations clear. The overall theme of the paper is to demonstrate, using known generative functions, the effects of ignoring the speed offset term when decoding velocity. We then show these effects in actual arm movement data. New decoders, Direct Regression and the ANN (plus full-OLE in supplemental material), were then shown to counteract these deficiencies. In the new BCI results, we show that both new decoders corrected most of the problems with the current OLE decoder and that the new ANN decoder worked especially well.

The comparison of OLE to direct regression in fig. 7 is a bit unfair/misleading given the substantial difference in the total number of units contributing to each. A matched population decoding comparison must be provided:

We now use the same group of units for each decoder. However the number of units differs between monkeys. We also clarify how the statistics were calculated using 10-fold cross-validation.

“Interestingly, when the excitability...” paragraph should also mention that recent results from Perel et al. (J Neurophysiol 2015) suggest threshold-crossings and multi-unit data seem to contain significant speed information:

We agree that the Perel et al paper would have supported the argument put forth in the paragraph, but we deleted the paragraph from this version of the manuscript to streamline the discussion.

Reviewer #3

My only main critical comment is in regard to the distortion effect of the constant b_0 before and after the movement for the offset model which is illustrated in Fig. 4 C, D, and E, as well as in Fig. 5N, Fig. 6C, Fig. 7C, and I. The mis-estimated baseline (b_0) (p. 9, line 302) is the result of using movement-related activity in the regression function. Since the change of activity of many neurons as a function of direction is not symmetrical around their baseline activity (Fig. 3B and C, in contrast to 3A), the offset (b_0) found using movement-related activity will be necessarily different than their actual baseline. A simple way to deal with this is to compute the regression using neuronal activity minus the recorded baseline. So, for example Eq. 4 would become:

$$y_i(t) - b_{0_i} = m_i |V(t)| \cos(\theta - \theta_{PDi}) + b_{si} |V(t)| + \epsilon_i$$

with b_{0_i} being the activity recorded during the baseline period (rather than a regression parameter). In this case, the baseline would necessarily correspond to the correct baseline and reduce the distortion of decoded kinematics before and after the movement period. However, the distortion in the decoded movement trajectory, as seen in some conditions, would remain essentially the same as illustrated in the article since that distortion has a different cause (as mentioned above):

The reviewer brought up a valid point regarding how the mis-estimated baseline could be addressed using an empirical measure, such as a ‘spontaneous rate’ found as baseline activity. We added this to the manuscript as a possible solution, but cautioned that while this might work well in laboratory paradigms such as the center-out task, its applicability in a more general setting may be difficult, as the meaning of ‘spontaneous’ becomes more nebulous. The underlying idea put forth by the reviewer is that it would be appropriate to find a baseline in the ‘no-movement’ context, so that speed (with a value of zero) could be extracted correctly in this condition. This is true, however one must then assume that this estimate is equally valid during movement (or in all the other contexts of real life). This becomes an interesting, and almost philosophical issue that can only be addressed with future experiments in which subjects use BCI in a wide range of conditions outside the laboratory.

Minor comments:

p. 2. The text (line 65) indicates that the average trajectories plotted in Fig. 1A were for Monkeys N and C, whereas the legend of the figure indicates that they were only for Monkey N. Please reconcile:

The figure has been replaced. The text and legend now correctly refer to both monkeys

p. 3. Line 74. "We recorded single neuron responses: 86 for Monkey N and 93 for Monkey C." please indicate how single neurons were isolated. (Incidentally, it would be interesting to know how important for decoding is it to isolate activity of single neurons relative to using multiunit activity. This is not the object of this study, but, if the authors find it appropriate to include it in the text, it would be interesting to know what their position on this is.):

The methods now clearly describe the isolation procedures for each monkey/experiment. For the arm movement data, single units were isolated. A mix of single and multiunit isolation was used in the BCI experiments. This is now described explicitly in the manuscript Methods: Neural Recording for the arm-movement data and Decoder Calibration for Closed-loop Control for the BCI recordings. While the idea that speed is preferentially represented in signals that have been spatially integrated (Moran and Schwartz, 1999, Perel et al, 2015) is interesting, we feel that this consideration is beyond the scope of the manuscript.

p. 3. Lines 93-95. I understand the need for resampling, but I am not sure what precisely is meant by "to match each bin with the velocity values adjusted by bin width":

We meant that by changing the bin width, the ration of distance/time (speed) changes, so we took that into account when re-sampling. This is now clarified.

p. 4. Since a time-dependent direction of movement $D(t)$ was part of Eq. 2, then its vector formulation in Eq. 3 should also have a time-dependent variable, that is, $\theta(t)$. Same comment for Eq. 5, 7, 9, and 10. However, since movement direction was approximately constant from the center to the target, then t could be omitted in Eq. 2 (and following Eqs.), and mention that movement direction was considered to be straight and t was omitted here for simplicity:

The equations have now been modified to include the time-dependency of direction. $|\mathbf{v}(t)|$ has been corrected as a term for speed.

p. 8. Line 261. "...poison noise..." should be corrected with "...Poisson noise...":

"Poison" has now been corrected to Poisson

p. 8. Eqs. 23, 24 The regressions were computed using firing data averaged over a time-window of interest, presumably between movement onset and offset; however, please specify the time window used for the regressions:

This encompassed all the data in the simulation. We now state this explicitly.

p. 13. Line 386. The central tendency (red arrow) of the von Mises fit in Fig. 6A is at negative 147 deg. Please correct the sign in the text:

We replaced this figure with Fig. 7 in the revision. We now just state that both distributions are skewed and indicate the central tendency with a red line in the figure.

Fig. 7. The thick gray line that represents average speed in the C, E, I, and K plots should be mentioned in the legend:

Thick gray line for actual average speed profile: This is defined in the Fig. 5 legend and now says that it applies to subsequent figures.

Reviewer #4

Concern 1: Novelty: The Authors propose an alternative extraction algorithm that bypasses explicit parameter encoding", the alternative method is a simple linear regression. The first thing any engineer would try on a new prediction problem; The paper presents the PVA and OLE methods for predicting movement velocity. These methods have become the de-facto standard in physiology texts, explaining how movement is related to firing rates of neurons in the motor cortex and how this relation can be used to estimate velocity. They suggest that conventional methods that make use of fitted neuronal rate models are biased in the general case of non-uniform neuronal distributions, and when the firing rate is modulated with respect to the magnitude of the velocity. This is, WE believe true but not surprising. Given the popularity of the PVA model (~2000 citations of the 1982 Georgopoulos paper and ~650 citations of the Moran & Schwarz paper) and the numerous computational studies which used this model to predict movements.

The authors could clarify more vividly the contribution of current analyses add to our current understanding by addressing some of the suggestions below.

Yet, as stated above, the notion of using alternative extraction algorithm that bypasses explicit parameter encoding, is attractive and will be hopefully used in new ways to demonstrate novelty and a new understanding:

A new hybrid ANN decoder was developed as a decoder that combines an initial encoding step with a simple artificial neural network. We performed an additional experiment to show how this decoder compared to the Direct Regression and to the OLE decoders during brain-control.

We wonder if the suggested algorithms represent the optimal approach to bypasses explicit parameter encoding. At this stage, contemporary science novel methods and algorithms are being developed in other regimes of machine learning and advanced controllers. Neuroscience will be advanced by applications of more visionary algorithms and test them against the linear regression approach. Algorithms that could be used to facilitate understanding of the motor cortex may include dynamic-adaptive controllers that can examine how past observations predict the future ones (like the brain) and adapt rapidly (like the brain) to create updated control of actions. Specifically, the paper and its message could become more attractive by modelling of the causal relationships in the brain-behaviour loop, based on the notion of adaptive-predictive features of internal models of the brain. However, the data used here is probably not optimal for this purpose:

While we agree that modeling causal relations in the brain-behavior loop is very interesting, this is well beyond the scope of this paper. Here, we are taking the rather modest approach of identifying a problem with current decoders. This problem stems from the observation that motor cortical firing rate during reaching should account for speed with both a gain and an offset term. Failure to do so, leads to characteristic decoding errors. Thus this paper stresses the importance of considering empirical encoding when designing decoders, with the modest goal of improving brain-controlled interfaces. We make no claims here of expanding scientific theory.

Concern 3: The paper proposes to put aside process-based mechanistic models in favor of standard (even non-regularized) linear regression from firing rates to instantaneous velocity (direct regression). The proposed solution (at this stage and version of the paper) is not yet convincing and could be a non-optimal compromise. It is non-mechanistic, providing no new insights on how the motor cortex is involved in motor control and probably too simplistic from the perspective of a BMI engineer (yet to be tested). Previous studies used of the Kalman filter (Wu and Donoghue) and kernel-based regressive models (KARMA, Shpigelman et al.) have shown how methods that also model movement dynamics and allow for inter-dependency between neuronal firing rates to affect the prediction of movement can out-perform simple linear regression. These and other approaches should be compared in the unbiased and extensive manner to the methods tested here. In sum, a critical comparison of direct-linear regression to other algorithms is insufficient at this stage of the paper. We can only guess that algorithms, which optimize discovery of causal relations between brain activity and behavior, would have to focus on the adaptive, predictive nature of the brain as a controller that dynamically processes, represents and executes the desired actions like movements with desired velocity profiles.

As mentioned above, we now directly compare Direct Regression to OLE and the hybrid ANN in actual BCI experiments.

The implication of the discussion (lines 488-502) that algorithms like Kalman-filter, which "rely on fitting encoding models to the firing rates of each unit in the recorded sample", are suffering from problems related to difficulty in inverting the models, is misguided. The difficulty is a specific feature of the PVA and OLE models. However, the direct regression (and Kalman filter or KARMA) do not suffer from this separation of model fitting criterion and velocity estimation error (they are fitted to minimize the error in velocity estimation). The 'problem' with the direct method is that it should not be interpreted, without validation, as providing neuronal tuning (because parameters that refer to one neuron would end up different if they were fitted with or without observation of the other neurons. If the authors think otherwise, we suggest that they try and make their case empirically (for example, by comparing the models on a test set not used for fitting the models (see other concerns, item 1 below):

The statement that Kalman Filters rely on encoding models is misguided: Here we disagree. The Kalman filter uses an observation model which is the same as an encoding model. This is at least "implicitly" inverted. However, as discussed for the full OLE, modeled residuals may be cancelled out, as most Kalman filters use the full covariance matrix to weight the contributions from individual units. We were careful to say that encoding inversion may be explicit or implicit.

There are several ways of treating the problem of biased decoding by OLE and PVA, which are sub-optimally fitted for their eventual use. One solution, as proposed by the authors, is to

abandon the PVA and OLE as presented in this study and opt for a direct linear regression method. As explained above (concern 3), may be a problematic solution.

Another way would be to find a method of fitting the OLE and PVA parameters so that they directly minimize the decoding error. This would assume that the models represent a mechanistic process by which the neuronal activity is causing the end-point velocity (through some presumed kinematics to hand dynamics transformation).

Yet another approach would be to encompass the assumptions of what neurons encode (how their firing rates are affected by movement velocity) into a probabilistic generative model of firing rates and find ways to fit them in the probabilistic modelling framework so that the resulting model can be used to estimate the underlying hidden ("unobserved") velocity variables from the observed firing rates. This, for example, is the approach taken by the Kalman filter model, which assumes linear state dynamics with Gaussian noise (the modeler can decide what to represent by the state E.g. the state can be the current position, velocity and acceleration) and a similar linear-Gaussian observation model:

This was a little confusing. Assuming that this refers to overfitting, we now use 10-fold cross validation to describe that variability of our results.

A probabilistic modelling approach that might produce a better decoding method than the described PVA method while still proposing a relation between intended movement and cortical activity and can be physiologically motivated.

Finally, other approaches can apply new neural networks (NN) models, with an attempt to feature the local circuits in cortex and predict how neuronal state changes when instructed to choose velocity in a given context. While NN suffered a long era of "depression" in previous years, new advances suggest that biologically-relevant networks can be used to find new ways to study the fundamental problem of how does neuronal population is modulated during control of movements:

While the OLE may be described as mathematically optimal in that it minimizes mean-squared error, in reality this is only true if a number of assumptions are met (e.g. linearity). Obviously we cannot prove the ANN is optimal. However, as with most applied methods, we compare several decoding approaches and show their relative performance. We believe that this demonstration shows convincingly that particular decoders perform better because they can handle firing-rate data that conforms to a speed encoding model comprised of gain and offset terms. Again, the scope of the paper is limited to this statement and we make no claims as to "finding biologically-relevant networks..." which admittedly is an interesting scientific problem.

Minor comments

1. Some of the notation is confusing. Most papers use single characters to represent variables. Non-capitalized bold characters are used for vectors and capitalized characters (typically bold) are only used for matrices.

Equations and notations were updated to follow convention.

2. We don't think the paper says how the PVA and the OLE parameters are estimated from recorded movements. (WE could have missed it?). We guess they are estimated to minimize

the mean squared error per unit but this is not stated. This information is important to understanding the claims of the study.

We estimated model parameters for each unit from recorded movement and firing rate data by minimizing the mean squared error. This is now explained in the Methods section.

3. figure 2: were offset ratios calculated for the best fitted time lag? Is the best fit calculated for each time lag and then the best fitting result kept? This is not clear:

For each unit, we fit the encoding model with different time lags (from -120 ms to 270 ms at 30 ms steps) between kinematics and firing rate data. The optimal time lag with the best fit was found for each unit. The offset ratio of a unit was then calculated using model parameters fit at its optimal time lag. Fig. 2c presents the histogram of the offset ratios from all units, and we report the median of the offset ratios in text. We revised the text and Fig. 2 legend to clarify this.

4. It is not clear if the authors chose to use stretched/ compresses time bins (so that movements are time aligned) because of their previous experience or did they test the possibility that these variations are relatively small and the movements are relatively uniform in their time/speed profiles). If this could have been avoided, trade-offing simpler descriptions for the price for some complexity in coding the data processing:

This was done to make movements time-aligned for ease of presentation and calculation when averaging across trials. Center-out arm movements used in this study were stereotyped. The variations in speed profiles were relatively small, as evidenced by the small standard errors of bin widths (see Methods, Data preprocessing). When presenting BCI results, trajectories and speed profiles were similarly aligned and resampled (Fig. 8). However, we did not adjust speed by bin width to preserve its original magnitude. We explained this in Methods and the Fig. 8 legend.

Reviewer #2:

Remarks to the Author:

Inoue and colleagues have significantly improved the manuscript from the initial submission. I thank the authors for thoughtfully incorporating feedback from the reviewers. The argumentation is more streamlined and the additional online BCI data further supports their claims. The analysis and presentation of the new online BCI data, however, needs improvements before the paper is suitable for publication.

Specifically, the quantification of online performance differences between the three different decoders can be improved. The authors primarily present videos and qualitative comparisons of performance ("it took the monkey extra effort," control "seemed effortless", and "straighter"). While I agree that the videos help convey a gestalt-like difference in the performance, the videos alone are insufficient for a comparison of performance. The speed during hold comparison is a useful comparison. Similar metrics of trajectory statistics (to quantify straightness) are also needed. Such metrics focused around the center and targets, I suspect, would be useful for quantifying the differences in control that reflect compensatory strategies due to inaccurate speed decoding. Behavioral measures like hold error rates would similarly be informative (though in and of themselves are limited metrics since they depend strongly on task structure). Finally, I would suggest that quantification of differences should be done as paired statistical tests with your within-day matched pairs. It's unclear from the current text how the statistical comparison of speed was done. Given that you tested decoders with two decoders per day in blocks, it is most accurate to compute performance comparisons between matched pairs within days. This controls for behavioral variability due to motivation levels, recorded units, etc. which fluctuate from day to day in experiments.

Specific and minor notes:

- It's quite striking how much larger/pronounced the OLE, DR, and ANN errors are in open-loop predictions versus closed-loop BCI performance. While not surprising based on the literature, this should be noted and incorporated into the discussion. It highlights both that some offline decoding errors are not necessarily always applicable to the closed-loop setting (as previously noted by many groups). But also makes the authors' observations that speed decoding errors do persist and influence control even in the closed-loop setting more remarkable. Deeper discussion of this point could greatly improve the manuscript.
- I find the presentation/discussion of the "full OLE" data somewhat strange. Having supplementary figures that are not supporting data presented in the results section, and are only discussed in the "discussion" section is pretty atypical. I would suggest revisiting the manuscript narrative to better incorporate any points you wish to make about these decoders into the results portion of the manuscript.
- The abstract should be modified to include a more precise quantification of bci performance differences
- Is there temporal binning/smoothing done on the speed trajectories presented in Fig 1? The data have oddly flat/linear segments that seem to suggest coarse binning.
- Text in lines 232 – 234 should be updated to make it more clear this uses a non-uniform, offset model simulation. This information is only included in the figure legend.
- Line 312, please revise to increase clarity. I assume those numbers are the OLE, DR, and ANN "respectively"?
- Line 339, I find the construction of this sentence a bit unclear. It also incorporates agrammatical phrases like "address for". Please revise.
- Line 410, "...a monkey learned rapidly to move..." The manuscript as currently presented does not provide any evidence or data on learning or rates of learning during online control. Please revise this discussion or amend the presentation to avoid unsupported claims.
- Please provide additional information on how videos were constructed and/or the structure of the BCI

task. Is the BCI cursor reset to the center after completion of a trial, or is the video edited to only show select trial snippets? Information on how time intervals for each video were chosen should also be included.

Reviewer #3:

Remarks to the Author:

The authors took into account all the concerns that I had raised, and modified the manuscript to my satisfaction.

Minor comment:

The word 'direction' is missing at the end of the legend of Fig. 4e.

Reviewer #4:

Remarks to the Author:

The paper is significantly improved with the addition of the hybrid ANN detector. However, the authors did not address some of our main concerns and I can see why. Citing from the authors' response: "While we agree that modeling causal relations in the brain-behavior loop is very interesting, this is well beyond the scope of this paper. Here, we are taking the rather modest approach of identifying a problem with current decoders"

I agree with the authors and accept their statement that this paper looks for a simple solution of speed representation with the current decoders, and they do it well. The paper has no flaws. We may agree less about the choice of the general approach and the tools. The paper merits publication in a peer-review journal and become available to the community of motor control.

Reply to the reviewers' comments:

We have fully addressed all of the comments and modified the paper to fully comply with the suggestions. Most importantly we have analyzed the BCI results as requested and added a figure to show pairwise daily comparisons of the decoders as requested by Reviewer #2.

Reviewer #2 (Remarks to the Author):

Inoue and colleagues have significantly improved the manuscript from the initial submission. I thank the authors for thoughtfully incorporating feedback from the reviewers. The argumentation is more streamlined and the additional online BCI data further supports their claims. The analysis and presentation of the new online BCI data, however, needs improvements before the paper is suitable for publication.

Specifically, the quantification of online performance differences between the three different decoders can be improved. The authors primarily present videos and qualitative comparisons of performance ("it took the monkey extra effort," control "seemed effortless", and "straighter"). While I agree that the videos help convey a gestalt-like difference in the performance, the videos alone are insufficient for a comparison of performance. The speed during hold comparison is a useful comparison. Similar metrics of trajectory statistics (to quantify straightness) are also needed. Such metrics focused around the center and targets, I suspect, would be useful for quantifying the differences in control that reflect compensatory strategies due to inaccurate speed decoding. Behavioral measures like hold error rates would similarly be informative (though in and of themselves are limited metrics since they depend strongly on task structure). Finally, I would suggest that quantification of differences should be done as paired statistical tests with your within-day matched pairs. It's unclear from the current text how the statistical comparison of speed was done. Given that you tested decoders with two decoders per day in blocks, it is most accurate to compute performance comparisons between matched pairs within days. This controls for behavioral variability due to motivation levels, recorded units, etc. which fluctuate from day to day in experiments.

Quantitative results (including rate of motion before and after the movement, target-hold success rate, and curvature of movement trajectory) and statistical comparisons between within-day decoder pairs were added as the new Figure 9. Main text was updated accordingly. Statistical comparison of speed based on multi-day data in the previous version of the manuscript was removed.

Specific and minor notes:

-It's quite striking how much larger/pronounced the OLE, DR, and ANN errors are in open-loop predictions versus closed-loop BCI performance. While not surprising based on the literature, this should be noted and incorporated into the discussion. It highlights both that some offline decoding errors are not necessarily always applicable to the closed-loop setting (as previously noted by many groups). But also makes the authors' observations that speed decoding errors do persist and influence control even in the closed-loop setting more remarkable. Deeper discussion of this point could greatly improve the manuscript.

The last few paragraphs of the discussion have been revised extensively. Lines 391-404 now read: *Careful examination of movement parameter encoding in simulations and "offline" actual*

movement data is useful for establishing models and testing putative decoding algorithms. However, actual testing of decoder performance in a BCI context is critical because different control strategies are likely used, leading to unforeseen changes in performance. When directional tuning functions are calculated as monkeys perform center-out arm movements and then recalculated as the same task is carried out in virtual reality using a BCI, the preferred directions of most neurons change^{13,20}. There may be multiple reasons for the changes in tuning between the BCI and arm-movement conditions. One set of explanations is based on considerations of the physical plant. A lack of somatic sensory input³³ and/or the non-activation of muscles during BCI³⁴ may be responsible for changes in the tuning function. Open-closed loop changes in the tuning function may also be due to short- and long-term learning that takes place in the BCI paradigm^{28,35,36} as the subject overcomes observed errors in the decoded movement. We tested pairs of decoders in daily experimental sessions with a monkey performing a brain-controlled center-out task (Figs. 8, 9).

-I find the presentation/discussion of the “full OLE” data somewhat strange. Having supplementary figures that are not supporting data presented in the results section, and are only discussed in the “discussion” section is pretty atypical. I would suggest revisiting the manuscript narrative to better incorporate any points you wish to make about these decoders into the results portion of the manuscript.

We have now removed reference and results related to the full OLE except in the discussion of how consideration of noise covariance can play a role in decoding. The two supplemental figures showing OLE have been removed. We felt that the results garnered using other OLE versions applied to the offline data were similar enough to the full OLE that it wasn't necessary to include these figures.

-The abstract should be modified to include a more precise quantification of bci performance differences

The abstract was modified to summarize the comparisons of BCI performance based on newly added quantitative performance metrics.

-Is there temporal binning/smoothing done on the speed trajectories presented in Fig 1? The data have oddly flat/linear segments that seem to suggest coarse binning.

There is no temporal binning/smoothing done on the speed profiles presented in Fig. 1. However, as described in Methods, single-trial speed profiles were temporally aligned, spline-interpolated, resampled, and averaged across all trials to generate the plots in Fig. 1. These data processing steps contributed to the smoothness of those speed profiles.

-Text in lines 232 – 234 should be updated to make it more clear this uses a non-uniform, offset model simulation. This information is only included in the figure legend.

We modified the text accordingly.

-Line 312, please revise to increase clarity. I assume those numbers are the OLE, DR, and ANN “respectively”?

Correct. However, these numbers were removed, as they were for data grouped across days which were replaced by comparisons between within-day datasets.

-Line 339, I find the construction of this sentence a bit unclear. It also incorporates agrammatical phrases like “address for”. Please revise.

This sentence was revised to improve clarity.

-Line 410, “...a monkey learned rapidly to move...” The manuscript as currently presented does not provide any evidence or data on learning or rates of learning during online control. Please revise this discussion or amend the presentation to avoid unsupported claims.

This sentence was revised to avoid any claims about learning rate.

-Please provide additional information on how videos were constructed and/or the structure of the BCI task. Is the BCI cursor reset to the center after completion of a trial, or is the video edited to only show select trial snippets? Information on how time intervals for each video were chosen should also be included.

It is now explained in Methods that the cursor was reset to the center at the start of each trials. This was also added to Supplementary Movie 1 legend. Supplementary Movie 1 was made with data chosen from one of those blocks using the OLE decoder that had relatively few failed trials (~20% of blocks using OLE). This block was also chosen to demonstrate both center-hold and target-hold errors. Supplementary Movie 2 used data of one block from those without failed trials, which were ~20% of all blocks recorded with the Direct Regression decoder. Similarly, Supplementary Movie 3 was also one of those blocks with no failed trials, which were ~40% of all blocks recorded with the ANN decoder. These were explained in movie legends.

Reviewer #3 (Remarks to the Author):

The authors took into account all the concerns that I had raised, and modified the manuscript to my satisfaction.

Minor comment:

The word 'direction' is missing at the end of the legend of Fig. 4e.

Fixed.

Reviewers' Comments:

Reviewer #2:

Remarks to the Author:

The authors have nicely addressed my comments. I'd like to congratulate them on the excellent manuscript.

Reply to the reviewers' comments:

Reviewer #2 (Remarks to the Author):

The authors have nicely addressed my comments. I'd like to congratulate them on the excellent manuscript.

We thank the reviewer for previous constructive comments that guided the revision of the manuscript.